# Strong Hilbert space fragmentation via emergent quantum drums in two dimensions

Anwesha Chattopadhyay[1,2], Bhaskar Mukherjee[3],
Krishnendu Sengupta[1] and Arnab Sen[1*]

**1** School of Physical Sciences, Indian Association for the Cultivation of Science,
Jadavpur, Kolkata 700032, India
**2** Department of Physics, School of Mathematical Sciences, Ramakrishna Mission
Vivekananda Educational and Research Institute, Belur, Howrah 711202, India
**3** Department of Physics and Astronomy, University College London,
Gower Street, London WC1E 6BT, United Kingdom

⋆ tpars@iacs.res.in

## Abstract

We introduce a disorder-free model of $S = 1/2$ spins on the square lattice in a constrained Hilbert space where two up-spins are not allowed simultaneously on any two neighboring sites of the lattice. The interactions are given by ring-exchange terms on elementary plaquettes that conserve both the total magnetization as well as dipole moment. We show that this model provides a tractable example of strong Hilbert space fragmentation in two dimensions with typical initial states evading thermalization with respect to the full Hilbert space. Given any product state, the system can be decomposed into disjoint spatial regions made of edge and/or vertex sharing plaquettes that we dub as "quantum drums". These quantum drums come in many shapes and sizes and specifying the plaquettes that belong to a drum fixes its spectrum. The spectra of some small drums is calculated analytically. We study two bigger quasi-one-dimensional drums numerically, dubbed "wire" and a "junction of two wires" respectively. We find that these possess a chaotic spectrum but also support distinct families of quantum many-body scars that cause periodic revivals from different initial states. The wire is shown to be equivalent to the one-dimensional PXP chain with open boundaries, a paradigmatic model for quantum many-body scarring; while the junction of two wires represents a distinct constrained model.



# 1 Introduction

A generic isolated quantum system with many degrees of freedom is expected to "self-thermalize" as it evolves unitarily under the dynamics of its own Hamiltonian [1]. This implies that pure states obtained from the time evolution of different initial states that share the same energy density cannot be distinguished from each other at late times using only local probes. A microscopic justification for this self-thermalization is provided by the eigenstate thermalization hypothesis (ETH) [2–5] that posits that high-energy eigenstates of such systems appear locally thermal with the temperature being set by the energy density of the eigenstate.

Rapid progress in producing and manipulating well-isolated quantum simulators such as ultracold gases [6, 7], trapped ions [8], Rydberg atom arrays [9] and superconducting qubits [10] has made it possible to study thermalization and its violations in such platforms. In particular, the experimental observation of late-time coherent oscillations from certain simple high-energy initial states in a kinetically-constrained chain of 51 Rydberg atoms [11] generated great interest in understanding thermalization in interacting theories with constrained Hilbert spaces. The revivals reported in Ref. [11] were shown to arise due to the large overlap of some simple initial states with a small set of nonthermal high-energy eigenstates, dubbed quantum

many-body scars (QMBS) in Refs. [12,13], in an otherwise non-integrable PXP model [14,15] that served as the minimal model for the experiment.

Subsequent theoretical studies have shown a plethora of interesting non-ergodic behavior in various models with constrained Hilbert spaces, including Hamiltonian formulations of lattice gauge theories [16–20] that may be realizable on quantum simulators [21–23]. These include different varieties of QMBS [24–47], disorder-free localization [48–51] as well as a richer ergodicity-breaking paradigm dubbed Hilbert space fragmentation [52,53]. Such forms of ETH-violation are distinct from the breakdown of ETH due to many-body localization [54–56] where strong disorder plays a crucial role.

Systems with Hilbert space fragmentation [57–70] often feature multiple conservation laws [52,53] which severely restrict the mobility of excitations. In such cases, the Hilbert space can split into exponentially many dynamically disconnected *fragments*. These fragments cannot be distinguished by any obvious global symmetries of the Hamiltonian [52,53]. Such fragments can either be finite or infinite-dimensional in size in the thermodynamic limit and can show vastly different dynamical properties, such as integrability [59,67], disorder-free localization [48–51,58,62] or QMBS [66,68] though large fragments are expected to typically satisfy a Krylov-restricted version of ETH [57]. Both *weak* and *strong fragmentation* is known to exist in one-dimensional (1D) models [52,53], with the two cases distinguished by whether the fraction of eigenstates violating the ETH are a set of measure zero or not in the thermodynamic limit. Weakly fragmented systems are similar to systems with QMBS since both situations lead to weak ergodicity breaking where typical initial states still thermalize [71]. However, strongly fragmented systems present a distinct form of ergodicity breaking that is different from systems with QMBS.

In Ref. [52], 1D spin models with both global charge and dipole conservation laws were considered and it was argued that such dipole-conserving models should exhibit Hilbert space fragmentation in any dimension [52,53] (for examples of fragmentation without global dipole conservation, see Refs. [31,59,60,62,64,66,68,69]). One of the tell-tale signs of fragmentation in such models is an exponential number of completely inert states that form one- dimensional fragments on their own. While examples of both weak and strong fragmentation are known in one dimension, it is not clear whether global dipole conservation alone is sufficient to lead to strong fragmentation in higher dimensions. This extra conservation ensures that Hilbert space fragments of different sizes can be constructed by embedding suitable "active" regions into "inert" backgrounds and surrounding the "active" regions by "shielding" regions; the shielding region, however, turns out to be of the same size or bigger than the active region it isolates [53]. This makes it difficult to construct explicit examples of strong fragmentation in two or higher dimensions.

In this paper, we will construct a model that shows strong Hilbert space fragmentation in two dimensions by considering $S = 1/2$ spins (equivalently, hard-core bosons) on the square lattice with ring-exchange terms on elementary plaquettes that are consistent with total magnetization (equivalently, boson number) conservation as well as global dipole moment conservation. The important additional ingredient in the model is the presence of a kinematic constraint that no two nearest neighbor sites can have two up-spins (bosons) simultaneously. Similar models with ring-exchange and other competing terms, but without the additional hardcore constraints, are known to have interesting low-energy phases and transitions [72–74]. High-energy properties of the unconstrained model with only the ring-exchange terms were studied recently in Ref. [70] where it was realized that such terms imply subsystem symmetries associated with the conservation of magnetization along each column and row of the square lattice. This leads to global dipole conservation and consequently Hilbert space fragmentation. However, the precise nature of the fragmentation (weak or strong) could not be established for this unconstrained model in Ref. [70].

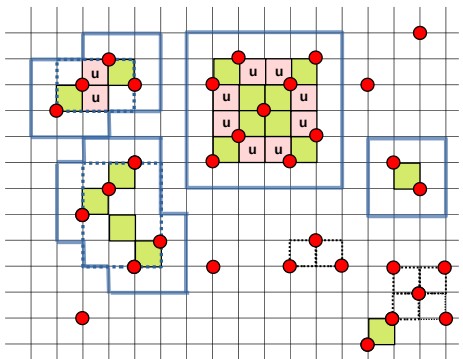

Figure 1: An initial state on the square lattice where the up-spins (bosons) are indicated in red while the other sites have down-spins (no bosons). The five quantum drums that correspond to this initial state are shown with their elementary plaquettes shaded. Plaquettes that are colored as green can have two up-spins (bosons) along both its diagonals during quantum evolution as explained in the text. The pink plaquettes, also labeled by "u", can have two up-spins (bosons) only along one of its two diagonals. Sites that do not belong to any of these five drums have inert up/down spins fixed by the initial condition. Each drum generates a separate fragment in Hilbert space with the corresponding fragment size being 3 for the top-left drum, 7 for the bottom-left drum, 24 for the middle drum and 2 for both the top-right and bottom-right drums. The boundary of the shielding region for 4 of the drums are shown using thick blue lines.

As we will show here, the enforcement of the kinematic constraints leads to several additional features, including strong ergodicity breaking and the emergence of "quantum drums", that were absent in the model considered in Ref. [70]. The quantum drums here can be viewed as the "active" regions which can then be surrounded by "shielding" regions of $O(1)$ thickness (in lattice units). Crucially, the thickness of the shielding regions does not grow with the size of the quantum drums. Each quantum drum is made of edge and/or vertex sharing elementary plaquettes and specifying the plaquettes that make a drum uniquely fixes its spectrum, thus justifying this particular nomenclature. We refer the reader to Fig. 1 for an example of quantum drums and their corresponding shielding regions that emerge from a particular initial state.

All the Hilbert space fragments of this model that are not one-dimensional, i.e., that do not correspond to inert Fock states, can be generated from a combination of appropriate quantum drums embedded in an otherwise inert background (which may itself shrink to zero for certain drums) (see Fig. 1). Thus, the Hilbert space can be decomposed as a direct sum over dynamically disconnected sectors that are completely labeled by quantum drums and any remaining inert spins that do not belong to a quantum drum. These quantum drums come in a variety of shapes and sizes and can be made of a finite number or an arbitrarily large number of plaquettes in the thermodynamic limit. Crucially, the nature of the Hilbert space fragments generated from large drums that form the largest Krylov subspaces and are, therefore, relevant for typical initial states allows for a proof of lack of thermalization by identifying either (a) an extensive number of single spin correlators or (b) an extensive number of next-nearest neighbor two-spin correlators whose expectation values stay pinned to their initial (non-thermal) values. To the best of our knowledge, this interacting theory provides the first example of *strong* Hilbert space fragmentation in two dimensions.

The rest of the paper is arranged as follows. In Sec. 2, we introduce the model and summarize some of its important properties. In Sec. 2.1, we discuss the quantum drums that emerge in this model in more detail. The classical construction of the drums, given an initial state, is explained in Sec. 2.1.1. The construction of the shielding regions of drums and closest approach of two drums such that these can still be considered independent of each other is explained in Sec. 2.1.2. In Sec. 2.1.3, a recursive procedure to generate bigger quantum drums starting from the most elementary one-plaquette drum is discussed. We introduce some particular drums, dubbed wires and junctions of wires and some other quasi-one dimensional (1D) and two-dimensional (2D) drums in Sec. 2.1.4. In Sec. 2.2, we give numerical evidence that the energy eigenvalues and their associated degeneracies from exact diagonalization (ED) on small systems can be completely understood in terms of the spectra of the quantum drums. We construct a large class of eigenstates with integer eigenvalues (including zero modes) from the packing of the simplest one-plaquette quantum drums in a macroscopic system size in Sec. 2.3. A wire decomposition of drums is introduced in Sec. 2.4, with Sec. 2.4.1 showing how to calculate fragment dimension and Sec. 2.4.2 showing how to construct entire drums from wire-decomposed reference states for some small drums. Evidence for strong Hilbert space fragmentation in this model is presented in Sec. 3. The numerical evidence from ED is presented in Sec. 3.1. In Sec. 3.2, we derive the scaling of the dimension of the Hilbert space fragments for the quantum drums composed of two long parallel wires to show the utility of the wire decomposition in obtaining the fragment size scaling for macroscopic drums. The wire decomposition allows us to derive the scaling of the dimension of the Hilbert space fragments associated with large 2D drums and determine which kinds of drums dominate statistically given a certain density of up-spins (bosons) and identify the Hilbert space fragment with the largest dimension in Sec. 3.3. We prove that typical initial states that belong to these large fragments (Krylov subspaces) *do not* thermalize with respect to the full Hilbert space in Sec. 3.3 by identifying either an extensive number of single-spin correlators or two-spin correlators that stay pinned to their initial non-thermal values. The analytical study for the spectra of certain small quantum drums is given in Sec. 4. A tree structure to represent the action of $H$ in the Fock space of a drum is explained in Sec. 4.1. The spectra of small wires is calculated in Sec. 4.2 while the spectra of other small quantum drums that can be viewed as building blocks of more complicated wire junctions is calculated in Sec. 4.3. The spectra of two different classes of bigger quasi-1D quantum drums, a wire and a particular junction of two wires, are addressed numerically using ED in Sec. 5. Both these large quantum drums can be interpreted as effective quasi-1D models with a spectrum that is symmetric around zero energy. A tree generating algorithm is described and the equivalence of the wire to the 1D PXP model on an open chain is shown in Sec. 5.1. The Hilbert space dimensions for both the drums are calculated analytically and level statistics are computed numerically in Sec. 5.2. The Hilbert space structure of the junction of two wires turns out to be completely different from that of the wire as discussed in both Sec. 5.1 and Sec. 5.2. One of these fragments is shown to have a macroscopically large number of exact zero modes while the other fragment has no zero modes in Sec. 5.3. Both fragments satisfy Krylov-restricted ETH but also support distinct families of QMBS that result in periodic revivals from different simple initial states as discussed in Sec. 5.4. Our numerical results for the wire show that open PXP chains of length $3n+1$, where $n$ is an integer, lead to enhanced fidelity revivals for the period-3 ordered initial $|\mathbb{Z}_3\rangle$ state without adding any optimal perturbations to the bare Hamiltonian; a feature which may have experimental consequence for Rydberg chains. The junction of two wires also shows QMBS and simple initial states from which clear revivals in fidelity are observed. Finally, we summarize our main results and conclude in Sec. 6.

## 2   Model and its properties

The Hamiltonian of the model is given by

$$H = J \sum_{j_x, j_y} \left( \sigma^+_{j_x, j_y} \sigma^+_{j_x+1, j_y+1} \sigma^-_{j_x+1, j_y} \sigma^-_{j_x, j_y+1} + \text{h.c.} \right), \tag{1}$$

where $\sigma^\alpha_{j_x, j_y}$ for $\alpha = x, y, z$ represent spin-half Pauli matrices at sites $(j_x, j_y)$ of a 2D square lattice, $\sigma^\pm_{j_x, j_y} = (\sigma^x_{j_x, j_y} \pm i \sigma^y_{j_x, j_y})/2$, and the lattice spacing has been set to unity ($a = 1$). The Hamiltonian is supplemented by the constraint that two up-spins can not occupy neighboring sites of the lattice; this is implemented by the operator relation

$$\left(1 + \sigma^z_{j_x, j_y}\right)\left(1 + \sigma^z_{j_x \pm 1, j_y}\right) = \left(1 + \sigma^z_{j_x, j_y}\right)\left(1 + \sigma^z_{j_x, j_y \pm 1}\right) = 0. \tag{2}$$

For finite $L_x \times L_y$ rectangular lattices, we will consider open boundary conditions and the constraint (Eq. 2) is then applied to the three/two nearest neighbors of $(j_x, j_y)$ for the edge/corner sites.

This system maps exactly to hard-core bosons with the following transformations:

$$2 b^\dagger_{j_x, j_y} b_{j_x, j_y} - 1 = \sigma^z_{j_x, j_y},$$
$$b^\dagger_{j_x, j_y} = \sigma^+_{j_x, j_y}, \tag{3}$$

where $b^\dagger_{j_x, j_y}$ is the boson creation operator and $n_{j_x, j_y} = b^\dagger_{j_x, j_y} b_{j_x, j_y}$ is the boson number operator at site $(j_x, j_y)$. For the rest of this work, we shall set $J = 1$. The terms in Eq. 1 can be viewed as ring-exchange terms on elementary plaquettes which convert a clockwise arrangement of $\sigma^z$ from being $(+1, -1, +1, -1)$ to $(-1, +1, -1, +1)$ (equivalently, an arrangement of bosons from $(1, 0, 1, 0)$ to $(0, 1, 0, 1)$) and vice-versa and annihilate other arrangements on a plaquette. It is convenient to define a vacuum state where all sites of the lattice have down-spins, i.e., no bosons for future reference. This model has the following properties:

- The many-body spectrum of $H$ is symmetric around the energy $E = 0$ for any finite $L_x \times L_y$ lattice with open boundary conditions (OBC). This is because the operator defined by

$$\mathcal{C} = \prod_{(j_x, j_y) \in (\text{even}, \text{even})} \sigma^z_{j_x, j_y}, \tag{4}$$

  satisfies $\{H, \mathcal{C}\} = 0$ where $\prod_{(j_x, j_y) \in (\text{even}, \text{even})}$ denotes a product over all the sites $(j_x, j_y)$ of the lattice such that both $j_x$ and $j_y$ are even. This implies that any many-body eigenstate of $H$ with an energy $E$ and denoted by $|E\rangle$ has a partner $\mathcal{C}|E\rangle$ that has the energy $-E$.

- Apart from discrete symmetries like rotations by $\pi/2$ (for $L_x = L_y$ lattices) and $\pi$ (for $L_x \neq L_y$ lattices), the model has a discrete reflection symmetry $\mathcal{R}$ where the axis of reflection can be taken to be the diagonal through $(0, 0)$ for $L_x = L_y$ or the perpendicular bisector of the longer side when $L_x \neq L_y$. $\mathcal{R}$ commutes with both the Hamiltonian $H$ and the "chirality" operator $\mathcal{C}$. This has the important consequence that the spectrum has exact zero modes whose number scales exponentially with the system size due to an index theorem shown in Ref. [75]. These zero modes are the only eigenstates of $H$ that also possess a definite "chiral charge" of $\pm 1$ under the action of $\mathcal{C}$.

- The model conserves the total magnetization (boson number) defined by $S^z_{\text{tot}} = \sum_{j_x, j_y} \sigma^z_{j_x, j_y}$.

  More interestingly, it conserves the following dipole moments in the $x$ and $y$ directions:

$$D_x = \sum_{j_x, j_y} j_x \sigma^z_{j_x, j_y}, \qquad D_y = \sum_{j_x, j_y} j_y \sigma^z_{j_x, j_y}. \tag{5}$$

This property follows from the fact that the total magnetization on each column and each row of the square lattice is separately conserved under the dynamics induced by $H$ (Eq. 1) as pointed out earlier in Ref. [70] in a similar model, but without the Hilbert space constraints. Models with the simultaneous conservation of total charge and dipole moment have been shown to have the property of Hilbert space fragmentation [52,53]. This model is also fragmented due to the same reasons.

- The simultaneous conservation of magnetization on each column and each row of the $L_x \times L_y$ lattice also implies disorder-free localization for a large class of initial states. To see this, let us consider the vacuum state on a $L_x \times L_y$ lattice with OBC and then create an excitation by flipping a subset of spins to $\sigma_j^z = +1$ such that the sites labelled by $j$ are contained inside or on the boundaries of a rectangle of finite extent smaller than the entire lattice. The aforementioned conservation property then ensures that these $\sigma^z = +1$ spins cannot be transported outside this bounding rectangle since all the rows/columns outside this region have their magnetizations pinned to their lowest possible value.

- This model has an exponentially large number (in system size) of zero modes that are simply inert states, i.e., Fock states in the computational basis that are annihilated by all the local terms in $H$, a property shared by other models that simultaneously conserve total charge and dipole moment. However, the constrained nature of the Hilbert space also leads to an an exponentially large number of *non-trivial* zero modes that emerge from Hilbert space fragments of various sizes larger than 1, ranging from 3 to $c^{L_x L_y}$, with $c > 1$, for $L_x, L_y \gg 1$.

- This model possesses eigenstates with exact non-zero integer eigenvalues when the Hamiltonian has the normalization of $J = 1$ in Eq. 1. Their number scales exponentially in $L_x L_y$ for integer eigenvalues ranging from $\pm 1$ to $\pm O(\sqrt{L_x L_y})$ for $L_x, L_y \gg 1$.

While we focus on the model Hamiltonian in Eq. 1 with $J = 1$ for the rest of the paper, it is useful to point out that that the fragmentation property stays *unchanged* even if $J$ is replaced by an arbitrary $J(j_x, j_y)$ and/or additional diagonal interactions (in the computational basis) are included. These only change the associated eigenvalues and eigenvectors but not the contributing Fock states in any of the Hilbert space fragments. The modified $H$ with an arbitrary $J(j_x, j_y)$ but no additional diagonal interactions still anticommutes with $\mathcal{C}$ (Eq. 4) which means that the many-body spectrum continues to have $E$ to $-E$ symmetry. While the trivial zero modes of $H$ in Eq. 1 (i.e., the inert states) persist for an arbitrary $J(j_x, j_y)$, the number of non-trivial zero modes decreases drastically due to the loss of the reflection symmetry $\mathcal{R}$, with fragments of sizes that are odd (even) integers contributing one (no) zero mode each. The presence of additional diagonal interactions destroy the $E$ to $-E$ symmetry of the many-body spectrum since $\mathcal{C}$ no longer anticommutes with the modified $H$.

## 2.1 Quantum drums

Due to the structure of $H$ (Eq. 1) and the nature of the constrained Hilbert space (Eq. 2), elementary plaquettes can have a maximum of two up-spins (bosons), along any one of the two diagonals, and these are the only local configurations that can have any dynamics. Furthermore, a plaquette with two up-spins (bosons) can influence the number of possible local configurations in neighboring two-spin plaquettes even if it can have the two up-spins (bosons) only along one of the diagonals but not the other due to kinematic constraints (Eq. 2). These two facts lead to the emergence of dynamically disconnected spatial structures called quantum drums on a $L_x \times L_y$ lattice with OBC.

To understand the origin of these drums, let us imagine a *classical* Markov process in which the *transition* from one Fock state to another is caused by a ring-exchange on some elementary plaquette with two up-spins (bosons). In the presence of the hard-core constraints specified in Eq. 2, the configuration space splits into mutually inaccessible fragments, with all configurations within a fragment being mutually accessible via some finite sequence of the allowed transitions. Crucially, each such fragment can be associated with a unique real-space structure composed of a collection of connected elementary plaquettes that share edges and/or vertices. The Hamiltonian $H$ (Eq. 1) acts in the space of mutually accessible configurations of each such fragment to generate the spectra of these quantum drums. From a dynamical point of view, the precise nature of the quantum drums is imprinted in the particular initial state that the system starts from.

We will specify two complementary construction procedures for quantum drums below, one which starts from a given product state in the computational basis (Sec. 2.1.1) and the other where the drums are constructed recursively starting from the most elementary one-plaquette drum (Sec. 2.1.3). Some of the important properties of quantum drums, which will be detailed out in the rest of the paper, are summarized below:

- Quantum drums are constructed of connected elementary plaquettes that share edges/vertices. A drum has no site that contains an inert up-spin (boson).

- All many-body eigenstates of $H$ (Eq. 1) can be expressed in terms of the tensor product of eigenstates of appropriate quantum drums and of the remaining inert (up/down) spins, if any, on sites that do not belong to any quantum drum. This point is illustrated in detail using ED results in Sec. 2.2.

- The spectrum of a drum is uniquely fixed once the plaquettes that belong to it are specified. The spectrum of any quantum drum is symmetric around $E = 0$. This follows from the above mentioned point and implies the existence of a corresponding chiral operator $\mathcal{C}_{\text{drum}}$ for each drum.

- A class of quasi-1D and 2D quantum drums have an internal reflection symmetry $\mathcal{R}_{\text{drum}}$ that commutes with both $\mathcal{C}_{\text{drum}}$ and $H$ resulting in an exponential number of exact zero modes as the size of the drum is increased.

- Any quantum drum conserves the total magnetization when only the spins (bosons) on the sites that belong to the drum are considered.

- Any quantum drum satisfies an internal subsystem symmetry of simultaneous conservation of magnetizations along each column and each row (where the column and row is defined with respect to the background $L_x \times L_y$ lattice) of the drum.

### 2.1.1 Constructing quantum drums associated with an initial product state

We first give a construction procedure that fixes all the quantum drums given a classical Fock state on a $L_x \times L_y$ lattice with OBC. An initial Fock state and its associated drums are shown in Fig. 1. The construction procedure is schematically shown in Fig. 2 for two drums starting from different Fock states. Given the Fock state, firstly all plaquettes with two up-spins (bosons) are shaded. Ring-exchange moves are attempted on such plaquettes to see whether any additional plaquettes with two up-spins (bosons) are generated which are again shaded. This process is repeated with the newly shaded plaquettes until no additional shaded plaquettes are generated. The shaded plaquettes are then subdivided into connected regions that comprise of elementary plaquettes that share edges and/or vertices. A final check has to be performed on each of these connected regions separately to construct the quantum drums.

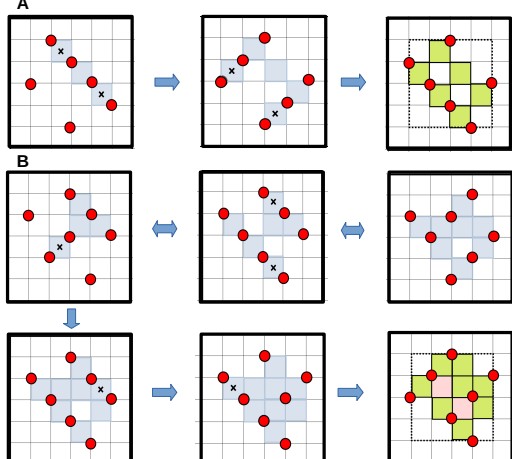

Figure 2: Illustration of the recursive construction of a quantum drum given two different initial states (marked by A and B in the figure) on a 7 × 7 lattice with OBC where the filled red dots indicate up-spins. The shaded plaquettes form part of a drum with the top-right and bottom-right panels indicating the drums for the initial states marked A and B respectively. A cross at the center of a plaquette indicates that a ring-exchange move is carried out for that plaquette. The green plaquettes in the top-right panel and the green and pink plaquettes in the bottom-right panel follow the same convention as used in Fig. 1.

If the mutually accessible Fock states from a connected region have certain sites where any up-spin (boson) remains the same in each of the configurations, these up-spins (bosons) are then labelled as inert and the shaded plaquettes containing any inert up-spins (bosons) are unshaded. The remaining shaded plaquettes that are still connected to each other via an edge or a vertex forms a quantum drum. This last check is necessary to rule out inert structures made entirely of plaquettes with two up-spins (bosons) [see Fig. 1 for an example composed of three up-spins (bosons) on two edge-sharing plaquettes] and to find spatial structures that can be decomposed into an inert region of up-spins (bosons) and a smaller quantum drum [see Fig. 1, bottom right for an example of such a decomposition].

Two simple examples of this construction are given for initial classical Fock states on a 7 × 7 lattice in Fig. 2 (top-left and middle-left panels), where the filled circles indicate up-spins (bosons) while the other sites have down-spins (no bosons). Let us first consider the top three panels. The initial state is given in the top-left panel marked as A and three plaquettes are shaded at this stage. Implementing ring-exchange moves on two of the shaded plaquettes indicated by crosses in the top-left panel generates two more shaded plaquettes as shown in top-middle panel. Implementing ring-exchange moves on the shaded plaquettes indicated by crosses in that panel generates another shaded plaquette in the top-right panel and further ring-exchanges do not generate any additional shaded plaquettes. The quantum drum generated by this initial state only contains elementary plaquettes that share vertices.

The initial state in the middle-left panel marked by B gives four shaded plaquettes. Implementing ring-exchange to this state for the plaquette indicated by a cross generates two more shaded plaquettes as shown in the following panel to the right. Carrying out ring-exchange moves on two more plaquettes as indicated by crosses generates two additional shaded plaquettes. To generate the other two shaded plaquettes that form the entire quantum drum, we go back to the initial Fock state shown in the bottom-left panel and perform two ring exchange moves on the plaquettes indicated by a cross one after the other as indicated in the bottom panels. The resulting quantum drum consists of only edge-sharing plaquettes in this case.

Both the quantum drums shown in Fig. 2 generate Hilbert space fragments of size 11 respectively, diagonalizing which results in the following eigenvalues:

$$\left(\pm\sqrt{\frac{1}{2}\left(9\pm\sqrt{57}\right)},\pm\sqrt{3},\pm\sqrt{2},0,0,0\right),$$
$$\left(\pm 2\sqrt{2},\pm\sqrt{3},-1,-1,+1,+1,0,0,0\right), \tag{6}$$

where the top (bottom) line in Eq. 6 refers to the eigenvalues for the quantum drum shown in the top-right (bottom-right) panel of Fig. 2. These two examples already illustrate that drums can have non-trivial zero modes, nonzero integer-valued eigenstates as well as eigenstates with irrational eigenvalues. We refer the reader to Sec. 4 for the explicit construction of the Hilbert space fragments associated with some small quantum drums.

### 2.1.2 Shielding region and closest approach of drums

Each quantum drum is associated with a shielding region of its own such that two quantum drums can fluctuate independently as long as the boundaries of their shielding regions do not cross. Given any Fock state consistent with a single quantum drum composed of a finite number of elementary plaquettes with the rest of the sites that do not belong to the drum being $\sigma^z = -1$ (no bosons), the corresponding shielding region can again be fixed by a classical construction. For this purpose, let us define

$$n_{\square_j} = \max\left[2 + (\sigma^z_{j_x,j_y} + \sigma^z_{j_x+1,j_y} + \sigma^z_{j_x,j_y+1} + \sigma^z_{j_x+1,j_y+1})/2\right], \tag{7}$$

where $n_{\square_j}$ is computed using *all* the Fock states that are accessible from the starting Fock state by ring-exchanges on elementary plaquettes. By definition (see Sec. 2.1.1), all plaquettes that belong to a quantum drum have $n_{\square_j} = 2$. For a quantum drum embedded in the vacuum state, all other elementary plaquettes must have either $n_{\square_j} = 0$ or 1. Importantly, only those plaquettes that lie to the exterior of the drum and directly share an edge or a vertex with the plaquettes on the perimeter of the drum *can* have $n_{\square_j} = 1$ while all other exterior plaquettes necessarily have $n_{\square_j} = 0$. To identify the subset of exterior plaquettes with $n_{\square_j} = 1$ requires constructing all the Fock states accessible to the given quantum drum since certain plaquettes with $n_{\square_j} = 2$ may allow two up-spins (bosons) only along one diagonal and not the other (e.g., see Fig. 1 for such plaquettes that are labeled by "u" and also indicated in pink.)

The shielding region of a quantum drum composed of a finite number of elementary plaquettes only consists of these exterior plaquettes with $n_{\square_j} = 1$ that directly share edges/vertices with the plaquettes on the perimeter of a quantum drum irrespective of the size of the drum. Thus, the thickness of the shielding region *does not* scale with the size of the quantum drum and remains $O(1)$ in lattice units (see Fig. 1 and Fig. 3 for examples). The boundary of the shielding region is defined as the closed curve formed by the edges that are common to the the exterior plaquettes with $n_{\square_j} = 1$ and $n_{\square_j} = 0$. The sites belonging to this boundary do not carry any $\sigma^z = +1$ spins (bosons).

Let us illustrate the construction of the shielding regions using two examples. First consider an elementary one-plaquette quantum drum starting from the vacuum state and then placing two $\sigma^z = +1$ spins (bosons) along any one of the diagonals of an elementary plaquette. Given this Fock state, ring-exchange is possible only on this elementary plaquette which then generates another Fock state where the $\sigma_z = +1$ spins (bosons) get transported to the other diagonal of this plaquette. Considering both these Fock states to compute $n_{\square_j}$ (Eq. 7) on each plaquette of the lattice, we see that $n_{\square_j} = 2$ for the flippable plaquette which is surrounded by $n_{\square_j} = 1$ and $n_{\square_j} = 0$ plaquettes, respectively (Fig. 3, left panel). The $n_{\square_j} = 2$ plaquette defines the quantum drum while the $n_{\square_j} = 1$ plaquettes along the perimeter of the

Figure 3: Two quantum drums are shown in the left and the right panels. The integers shown inside each plaquette refers to $n_{\square_j}$ given by Eq. 7 on each plaquette of the lattice using all the Fock states generated when the quantum drum is embedded in the vacuum state. The green (pink) plaquettes in the two quantum drums follow the same convention as used in Fig. 1. The boundary of the shielding region of both drums is shown using thick blue lines in both panels.

quantum drum define the shielding region associated with this drum. The shielding region terminates at the boundary of these $n_{\square_j} = 1$ and the $n_{\square_j} = 0$ plaquettes (Fig. 3, left panel). By construction, the sites at the boundary of the shielding region cannot have $\sigma^z = +1$ spins (bosons). A more complicated quantum drum is shown in Fig. 3, right panel which can be generated from the vacuum state by, e.g., placing two $\sigma^z = +1$ spins (bosons) each along parallel diagonals of the left-most and the right-most plaquette contained in the quantum drum such that the hard-core constraints are not violated. Performing all possible ring-exchanges for this quantum drum generates two more Fock states. Considering these three Fock states to compute $n_{\square_j}$ on each plaquette of the lattice, the four $n_{\square_j} = 2$ plaquettes, which are all connected to each other by edges for this particular drum, now define this bigger quantum drum (Fig. 3, right panel) while the $n_{\square_j} = 1$ plaquettes along the perimeter of the quantum drum define the shielding region as before (Fig. 3, right panel). The shielding region is more complicated compared to the one-plaquette drum and its boundary is again defined by the boundary of the $n_{\square_j} = 1$ and the $n_{\square_j} = 0$ plaquettes (Fig. 3, right panel). This classical construction procedure for the shielding region can be carried out for any arbitrary quantum drum composed of a finite number of elementary plaquettes.

We can now ask for the closest approach of any two quantum drums embedded in the vacuum state such that both the drums can be viewed to be independent of each other. Up-spins (bosons) cannot be transported outside the closed boundary of the shielding region of a quantum drum. Since the boundary sites do not carry any up-spin (boson), two quantum drums stay independent of each other as long as the boundaries of their corresponding shielding regions do not intersect; they may at most touch each other. For example, this is the case in Fig. 1 which explains why the different quantum drums can be considered to be independent of each other. When the boundaries of the shielding regions first cross each other, the corresponding quantum drums in their interior have to necessary change according to one of the following three possibilities: (i) the two quantum drums fuse to produce a bigger quantum drum, (ii) a spatial structure is produced such that it can be decomposed into an inert region of up-spins (bosons) and a smaller quantum drum, and (iii) a fully inert region of up-spins (bosons) is formed.

### 2.1.3 Recursive construction of bigger drums from smaller drums

We now present a complementary drum construction procedure to the one explained in Sec. 2.1.1 which does not need the specification of a product state on the entire $L_x \times L_y$ lattice. Instead, this recursive construction creates larger drums starting from smaller ones.

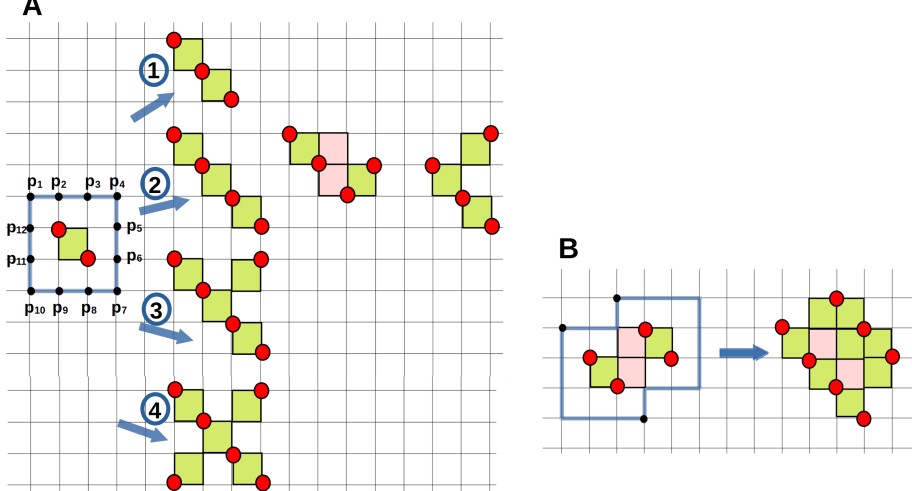

Figure 4: The left panel shows how adding up-spins (bosons) to the sites at the boundary of the shielding region (indicated by thick blue lines and also labelled from $p_1$ to $p_{12}$) of an elementary one-plaquette drum (marked as A) leads to a variety of larger quantum drums which are grouped according to the addition of $\textcircled{n}$ up-spins (bosons) where $n = 1, 2, 3, 4$. The right panel illustrates the same concept for a more complicated quantum drum (marked as B) where three up-spins (bosons) (indicated by black dots) are added to three sites of the boundary of the shielding region (indicated by thick blue lines). Filled red dots indicate up-spins (bosons). The green (pink) plaquettes in all the quantum drums follow the same convention as used in Fig. 1.

We start with a Fock state consistent with a particular quantum drum composed of certain connected plaquettes, where the sites that do not belong to the drum are assigned $\sigma^z = -1$ spins (no bosons). This is equivalent to embedding the quantum drum in the vacuum state. By definition, the boundary of the shielding region of such a drum has $\sigma^z = -1$ (no bosons). A natural way to construct bigger drums is to choose a subset of the sites that belong to this boundary of the shielding region and then replace $\sigma^z = -1$ by $\sigma^z = +1$ at these selected sites. This generates a new Fock state from which, using the procedure of identifying a drum from a Fock state explained in Sec. 2.1.1, one gets one of the following three possibilities: (i) a bigger quantum drum with no inert up-spins (bosons), (ii) a partially active structure that can be decomposed into a smaller quantum drum and a non-zero number of inert up-spins (bosons), and (iii) a completely frozen structure with all up-spins (bosons) being inert.

Two examples of this recursive construction to generate bigger drums starting from a smaller drum are shown in Fig. 4. We first start with a Fock state consistent with an elementary one-plaquette drum in the left panel of Fig. 4, marked as A. In this case, the boundary of the shielding region is a square that consists of twelve sites, labelled as $p_1, \cdots, p_{12}$ in Fig. 4. New Fock states, consistent with larger drums, can be created by adding one/two/three or four up-spins (bosons) in this boundary region as indicated by the groups labelled by $\textcircled{1}$, $\textcircled{2}$, $\textcircled{3}$ and $\textcircled{4}$ in Fig. 4, left panel. Adding a single up-spin (boson) at $p_1$ or $p_4$ generates Fock states consistent with a drum composed of two elementary plaquettes that share a vertex as shown in Fig. 4, left panel, group labelled by $\textcircled{1}$. Adding two up-spins (bosons) on the boundary of the shielding region in different ways leads leads to Fock states consistent with three different drums as shown in Fig. 4, left panel, group labelled by $\textcircled{2}$. E.g., adding up-spins (bosons) at $p_1$ and $p_7$ leads to a Fock state consistent with a drum with three plaquettes that share vertices

along a single diagonal (leftmost drum shown in the group labelled by ② in left panel of Fig. 4), at $p_1$ and $p_5$ leads to a Fock state consistent with a drum with four plaquettes that are connected by edges (middle drum shown in the group labelled by ② in left panel of Fig. 4), and at $p_4$ and $p_7$ leads to a Fock state consistent with a drum with three plaquettes that again share vertices, but not along a single diagonal (rightmost drum shown in the group labelled by ② in left panel of Fig. 4)). Adding three up-spins (bosons) at, e.g., $p_1$, $p_4$ and $p_7$, leads to a Fock state consistent with a quantum drum with four plaquettes that are connected by vertices as shown in the group labelled by ③ in left panel of Fig. 4. Finally, adding four up-spins (bosons) at $p_1$, $p_4$, $p_7$, and $p_{10}$ leads to a Fock state consistent with a quantum drum with five plaquettes connected by vertices (Fig. 4, left panel, group labelled by ④). To illustrate possibility (ii), we can add four up-spins (bosons) at $p_1$, $p_3$, $p_7$ and $p_{11}$ (Fig. 4, left panel) which leads to a Fock state consistent with a single-plaquette quantum drum containing sites $p_6$, $p_7$ and $p_8$ while the other up-spins (bosons) become inert. To illustrate possibility (iii), we can add a single up-spin (boson) at $p_5$ (Fig. 4, left panel) to generate a Fock state that has only inert up-spins (bosons).

This recursive procedure can be carried forth for the bigger quantum drums to produce more complicated quantum drums. An example is shown in panel marked as B (Fig. 4, right panel) where three up-spins (bosons), indicated by filled black dots, are placed on the boundary of the shielding region of a quantum drum, previously produced by adding two up-spins (bosons) at the boundary of the shielding region of the elementary single-plaquette drum, which leads to a bigger quantum drum with ten elementary plaquettes that are connected by edges. In principle, this recursive procedure can be used to generate and enumerate all possible quantum drums until a given stage of the recursion starting from the most elementary one-plaquette drum, but we leave this for a possible future investigation.

### 2.1.4 Wires, junctions of wires, other quasi-1D and 2D drums

As is already evident from the examples we have constructed so far, quantum drums come in several shapes and sizes, from being composed of a single elementary plaquette (Fig. 1) to a finite number of plaquettes (Fig. 1 and Fig. 2). One can even construct quantum drums with an arbitrarily large number of plaquettes in the thermodynamic limit. These varieties of drums can be quasi-1D or 2D in nature. We dub the *simplest* quasi-1D drum as a wire. A wire is composed of $N_p$ plaquettes that share vertices along a single diagonal and resemble straight wires (see Fig. 4, left panel for three such drums with $N_p = 1, 2, 3$). Such a wire can be constructed with any $N_p \geq 1$ that leads to a quasi-1D structure for $N_p \gg 1$.

Interestingly, one can create other quantum drums that resemble different kinds of junctions of such wires. Examples of such quantum drums are shown in Fig. 5. In the top panel, the quantum drums marked by A and B can be viewed as two different junctions of two wires, while in the bottom panel, the quantum drum marked by C (D) can be viewed as a junction of three (four) quantum wires. Wires can be used to build still more intricate quasi-1D as well as 2D drums (see Sec. 2.4 for details). The fragment sizes for large quasi-1D (2D) drums scale as $\alpha^l$ ($\beta^{l^2}$) where $\alpha > 1$ ($\beta > 1$) as $l \gg 1$ where $l$ represents the linear dimension of the drum and $\alpha$ ($\beta$) depend on the nature of the quantum drum under consideration. Each such quantum drum can be viewed as an interesting example of an interacting quasi-1D/2D model with a constrained Hilbert space that also satisfies an internal subsystem symmetry of simultaneous conservation of magnetizations along each column and each row of the drum, where the columns/rows are defined with respect to the $L_x \times L_y$ lattice in which the drum is embedded, when only the sites that belong to the drum are considered.

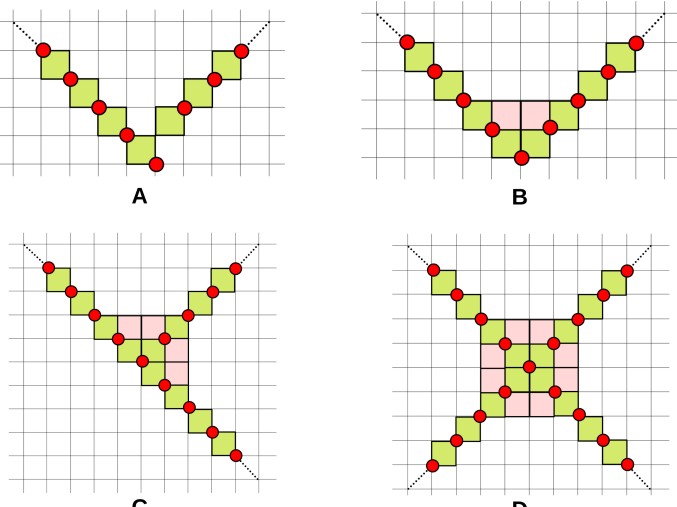

Figure 5: The quantum drums marked from A-D can be viewed as examples of different kinds of junctions of wires. A and B show examples of junctions of two wires while C (D) shows an example of a junction of three (four) wires. The green (pink) plaquettes in all the quantum drums follow the same convention as used in Fig. 1. The filled red dots indicate up-spins in all the panels and represent just one of the many possible Fock states of the corresponding drum.

## 2.2 Exact diagonalization of small lattices and deciphering spectrum using drums

The constrained nature of the Hilbert space reduces the number of allowed Fock states from $2^{L^2}$ to $\kappa^{L^2}$ where $\kappa \approx 1.503\cdots$ is the hard square entropy constant [76] for a square lattice with $L \gg 1$. This growth of the Hilbert space dimension with $L$ is, nonetheless, still too large to perform ED for the full spectrum for even moderately large values of $L$. However, analysing the numerical results for small $L \times L$ lattices is already instructive.

Let us first consider a $5 \times 5$ lattice and focus on the total magnetization sector with 5 up-spins (bosons). This gives a Hilbert space dimension of 10741 from direct enumeration taking the hard-core constraints in account. Plotting the histogram of the energy eigenvalues obtained from full ED reveals that the eigenvalues are clustered around only a *few* special values (up to machine precision) (Fig. 6, left panel) unlike what is expected of a generic interacting system with a similar Hilbert space dimension. Furthermore, while an explicit construction shows that there are 4559 inert Fock states that are trivially annihilated by $H$ (Eq. 1) in this magnetization sector, ED reveals that there are a total of 5525 zero modes (with zero eigenvalue within machine precision) implying the presence of 966 non-trivial zero modes. ED also shows the presence of 1580 eigenmodes with eigenvalue $+1$ ($-1$) and 196 eigenmodes with eigenvalue $+2$ ($-2$). Such non-zero integer eigenvalues are unexpected in generic interacting models which have highly irrational eigenvalues that cannot be expressed in any simple closed form. These and other features of the full ED data can be completely understood in terms of quantum drums (Fig. 6, right panel). The 10741-dimensional Hilbert space in the computational basis gets fragmented into 4559 (1-dimensional), 1552 (2-dimensional), 434 (3-dimensional), 324 (4-dimensional), 32 (5-dimensional), 32 (6-dimensional), 16 (7-dimensional) and 2 (8-dimensional) Hilbert space fragments. The 1-dimensional fragments simply correspond to the inert Fock states that are annihilated by all local terms of $H$ (Eq. 1) (and are denoted collectively by panel marked E in Fig. 6, right panel). All the other fragments

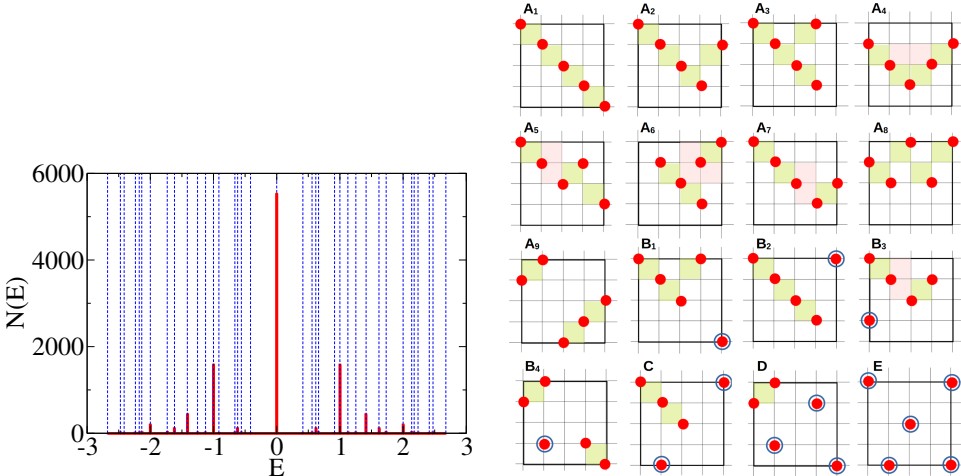

Figure 6: (Left panel) Histogram of the energy eigenvalues for a 5 × 5 lattice with 5 up-spins and OBC. The vertical dotted blue lines indicate the allowed eigenvalues from Eq. 8. (Right panel) The allowed quantum drums in this system where the up-spins (bosons) are indicated by filled red dots and the plaquettes that belong to quantum drums are shaded. The green (pink) plaquettes in all the quantum drums follow the same convention as used in Fig. 1. The inert up-spins (bosons) are indicated by a blue circle around the filled red dot. Each quantum drum is consistent with more than one Fock state with only one representative Fock state shown here. The different eigenstates can be viewed as modes of these quantum drums.

can be viewed as being generated from a collection of appropriate quantum drums and any remaining inert up-spins (bosons) that are not part of any drum.

Given the lattice dimensions and the number of up-spins (bosons), only certain drums are allowed with specific degeneracies set by the lattice. The different drum configurations along with one representative Fock state for each is shown in Fig. 6 (right panel) and are marked as $A_1, \cdots, A_9, B_1, \cdots, B_4$, C and D. The configurations labeled from $A_1$ to $A_8$ consist of different types of single drums that contains all the 5 up-spins (bosons) while $A_9$ consist of two independent drums, one with 2 up-spins (bosons) and another with 3 up-spins (bosons), respectively. The configurations $B_1$, $B_2$ and $B_3$ have a single quantum drum each with 4 up-spins (bosons) while 1 up-spin (boson) is inert as it does not belong to the drum. The configuration $B_4$ has two independent one-plaquette drums and 1 inert up-spin (boson) that does not belong to any of the 2 drums. Configuration C (D) has a drum with 3 (2) up-spins (bosons) and 2 (3) inert up-spins (bosons).

The degeneracies associated with the different configurations $A_1$ to E are indicated inside [ ] for each case in Eq. 8 and arise from the number of distinct ways in which the given drums and any inert up-spins (bosons) can be placed on the 5 × 5 lattice with OBC. For example, $A_1$ has a degeneracy of two because there are two diagonals along which the associated drum may be placed. Similarly, $A_2$ has a degeneracy of sixteen since there are sixteen distinct ways to place a "L" composed of four connected plaquettes that form the associated drum on this lattice. The other degeneracies given in Eq. 8 can be computed similarly. The spectrum shown for $A_1$ to E in Eq. 8 can be straightforwardly calculated by solving for the spectra of the constituent drums. The eigenspectra of all the fragments that arise from these quantum drums, barring the drum shown in $A_2$, can be expressed in closed form and show a variety of eigenvalues including zero modes, non-zero integer modes and irrational modes (Eq. 8). The extra non-trivial zero modes and their degeneracies can also be understood as zero modes of quantum drums shown

in $A_1$, $A_2$, $A_3$, $A_4$, $A_6$, $A_8$, $B_2$, $B_3$, $B_4$ and C (Eq. 8). It is useful to stress here that while certain drums, e.g., the one contained in $B_3$ and the one contained in C, are evidently different from each other, they have identical spectra (Eq. 8).

$$
\begin{aligned}
&A_1[2] \rightarrow \left( \pm\sqrt{4+\sqrt{10}}, \pm\sqrt{2}, \pm\sqrt{4-\sqrt{10}}, 0, 0 \right), \\
&A_2[16] \rightarrow (\pm 2.47367\cdots, \pm 1.25235\cdots, \pm 0.559107\cdots, 0), \\
&A_3[16] \rightarrow \left( \pm\sqrt{3+\sqrt{3}}, \pm\sqrt{3-\sqrt{3}}, 0, 0 \right), \\
&A_4[12] \rightarrow (\pm\sqrt{5}, \pm 1, 0, 0), \qquad A_5[8], A_7[8], B_1[96] \rightarrow \left( \pm\frac{1}{2}(1\pm\sqrt{5}) \right), \\
&A_6[16] \rightarrow (\pm\sqrt{3}, 0, 0), \qquad A_8[12] \rightarrow (\pm\sqrt{3}, \pm 1, 0), \\
&\qquad A_9[4] \rightarrow (\pm(1\pm\sqrt{2}), \pm 1), \qquad B_2[20] \rightarrow \left( \pm\sqrt{\frac{1}{2}(5\pm\sqrt{17})}, 0 \right), \\
&B_3[128], C[306] \rightarrow (\pm\sqrt{2}, 0), \qquad B_4[196] \rightarrow (\pm 2, 0, 0), \\
&\qquad D[1552] \rightarrow \pm 1, \qquad E[4559] \rightarrow 0.
\end{aligned}
\tag{8}
$$

## 2.3 Eigenstates with integer energies from packing of one-plaquette drums

Eigenstates composed of only elementary one-plaquette quantum drums and inert spins already generate non-trivial zero modes and non-zero integer eigenvalues. These can be viewed as the 2D generalization of bubble eigenstates discussed in a 1D model of Hilbert space fragmentation [66]. Hilbert space fragments with $n_0$ such independent one-plaquette drums have a dimensionality of $2^{n_0}$ since each such elementary quantum drum is consistent with two configurations on the plaquette. An extensive number of such elementary quantum drums are needed to form finite energy-density eigenstates of $H$ with a macroscopic number of up-spins (bosons) (Fig. 7). The closest packing of these elementary quantum drums such that the boundaries of their shielding regions do not overlap is shown in Fig. 7 which yields the maximum possible value of $n_0 = L^2/9$ for a $L \times L$ square lattice when $L \gg 1$ thus fixing the corresponding fragment's dimension to be equal to

$$
(2^{1/9})^{L^2} \approx (1.08006\cdots)^{L^2},
\tag{9}
$$

and the density of up-spins (bosons) to be $n = 2/9$. The corresponding matrix can be immediately diagonalized by noting that the form of $H$ projected to any $n_0 \neq 0$ fragment produced solely by elementary one-plaquette quantum drums equals

$$
H_{\text{eff}} = \sum_{i=1}^{n_0} \tau_i^x,
\tag{10}
$$

where $i$ denotes the center of an elementary drum plaquette, and $\tau_i^x$ locally flips an arrangement of $(+1, -1, +1, -1)$ to $(-1, +1, -1, +1)$ and vice-versa on that drum in the computational basis. This "non-interacting" $H_{\text{eff}}$ only leads to integer eigenvalues for any $n_0$. If $n$ $(n_0 - n)$ of the elementary quantum drums are associated with an eigenvalue $\tau_i^x = +1(-1)$, the resulting eigenstate has energy $E = 2n - n_0$. Clearly, there are $\binom{n_0}{n}$ distinct eigenstates that have the same energy $E = 2n - n_0$. Assuming that both $n_0, n \gg 1$, the degeneracy $\Omega(n)$ of such eigenstates is bounded below by

$$
\Omega(n) > 2^{n_0} \sqrt{\frac{2}{\pi n_0}} \exp\left( 2n_0 \left( x - \frac{1}{2} \right)^2 \right),
\tag{11}
$$

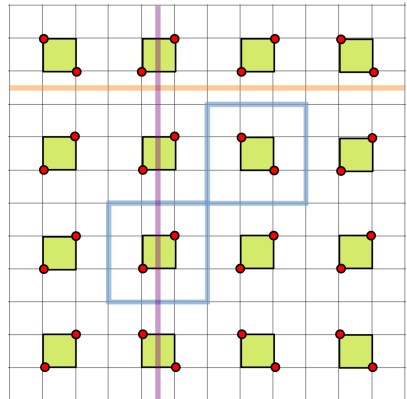

Figure 7: Close packing of elementary one-plaquette quantum drums shown, where the drum plaquettes are shaded, alongwith the boundaries of the shielding regions for two such drums (shown as thick blue lines). An initial Fock state which is consistent with this arrangement of quantum drums and where up-spins (bosons) are indicated by filled red dots is also shown. Two bipartition cuts are also shown as thick orange and purple lines.

where $n_0 = L^2/9$ for the largest such fragment (Fig. 7) and $x = n/n_0$. This bound immediately shows that the number of such integer eigenstates is exponentially large in the system size for integer energies that range from $E = 0$ to $|E| \sim O(L)$ (while the maximum value of the integer energy $|E| = L^2/9$ when $L \gg 1$ for a $L \times L$ square lattice with OBC). These high-energy eigenstates satisfy a strict area law scaling of entanglement entropy with the entanglement entropy of an arbitrary bipartition, $S_{\text{bp}} = bL$, where $b$ can range from 0 to $\ln(2)/3$ (examples of two such bipartition cuts which give the extreme values of $b$ are shown as thick lines in Fig. 7), depending on the nature of the bipartition.

Any Fock state consistent with $n_0$ independent one-plaquette drums (e.g., one such Fock state is shown in Fig. 7 where the red filled dots represent up-spins (bosons)) shows persistent oscillations with a time-period $T = \pi$ under unitary evolution under $H$ for a class of local operators. This can be directly related to the non-interacting nature of $H_{\text{eff}}$ in Eq. 10 which leads to the following *emergent* dynamical symmetry [77]:

$$\left[ P_{\text{eff}} H P_{\text{eff}}, \frac{\tau_i^y + i\tau_i^z}{2} \right] = \omega \left( \frac{\tau_i^y + i\tau_i^z}{2} \right), \tag{12}$$

where $P_{\text{eff}}$ is a projection operator to the Fock space with $n_0$ one-plaquette drums and $\omega = 2$ given the form of $H_{\text{eff}}$ in Eq. 10. Thus, for any such initial Fock state, any local operator with a finite overlap with any of the $\left( \frac{\tau_i^y + i\tau_i^z}{2} \right)$ operators will show persistent oscillations with a time period $T = 2\pi/\omega = \pi$.

## 2.4 Wire decomposition of quantum drums

As introduced earlier, wires represent the basic quantum drums that can be generated for any given number of plaquettes, $N_p$, by arranging them in a vertex-sharing pattern along any one of the two diagonal directions of the parent $L_x \times L_y$ lattice. A *reference Fock state* of the wire can be taken to be all the $N_p + 1$ up-spins (bosons) to be arranged along the length of the drum. The shielding region around a wire consists of all plaquettes that share either an edge or a vertex with any of the $N_p$ plaquettes that belong to the drum. We refer the reader to Sec. 4.2 (Sec. 5) where the spectrum for wires with small (large) $N_p$ shall be discussed. For now, it is

sufficient to note that the number of Fock states generated by a wire with $N_p$ plaquettes equals $F_{N_p+2}$ (a Fibonacci number) where $F_0 = 0$, $F_1 = 1$, and $F_n = F_{n-1} + F_{n-2}$ for $n > 1$ (see Sec. 5.2 for the derivation).

A reference Fock state for more complicated (non-wire) drums can be constructed from two or more parallel wires with differing lengths in general, with each wire being in its reference state, and any remaining unpaired up-spins (bosons) [that do not belong to any of the wires] that lie at a minimum distance of $(3/2)\sqrt{2}$ (in lattice units) from the nearest wire. Some of the Fock states displayed in Fig. 2 and Fig. 4 already serve to illustrate this concept of *wire-decomposed reference states*. The state in the top-middle panel (Fig. 2) can be viewed as a reference Fock state composed of two parallel wires with $N_p = 2$ each for the drum in the top-right panel (Fig. 2). Similarly, the state in the center-middle panel (Fig. 2) can be viewed as a reference Fock state composed of two parallel wires, one with $N_p = 2$ and another with $N_p = 3$, for the drum in the bottom-right panel (Fig. 2). The Fock states marked by ③ and ④ in Fig. 4 consist of unpaired up-spin(s) (boson(s)) at a distance of $(3/2)\sqrt{2}$ from a wire with $N_p = 3$. The parallel wires that make up a wire-decomposed reference state cannot exist as independent drums since either the shielding region of a wire overlaps with that of other parallel wires or unpaired up-spins (bosons) exist at the boundary of a shielding region of some wire(s).

In this section, we will see that

1. Drums composed of only vertex-sharing plaquettes can be built from a reference Fock state where the parallel wires can fluctuate *simultaneously* to access *all their internal states* without violating any of the hard-core constraints of the model. Such wires are separated by a distance of $2\sqrt{2}$ or greater from each other. All the Fock states of such drums can be generated from the fluctuations of these parallel wires and possibly, other sets of parallel wires in the same direction or perpendicular to the direction of the original set of wires (with all simultaneous fluctuations again allowed).

2. Drums composed of only edge-sharing plaquettes can be built from a reference Fock state where the parallel wires *cannot* fluctuate simultaneously to access all their internal states being at a distance of $(3/2)\sqrt{2}$ from each other, but only do so if alternate wires are kept in their reference state. The Fock states of such drums can be generated from the fluctuations of the alternate parallel wires and possibly, other sets of alternate parallel wires in the same direction or perpendicular to the original set of wires. However, not all simultaneous fluctuations of such consecutive wires are disallowed by the hard-core constraints of the model and these can be represented as *additional* excitations of elementary plaquettes that are separated by 3 lattice units along either $x$ or $y$, or both, such that these plaquettes can fluctuate *independently*.

3. Drums with both edge-sharing as well as vertex-sharing plaquettes can be built from a reference Fock state that consists of parallel wires such that while all the wires cannot fluctuate simultaneously to access all their individual states, some consecutive wires can do so if the other wires are kept in their reference state.

### 2.4.1  Calculating fragment dimension from wire decomposition

We first demonstrate the aforementioned concepts for small quantum drums before considering macroscopic quantum drums and their corresponding fragment sizes in later Sections (Sec. 3.2 and Sec. 3.3). We start with the simplest case of two $N_p = 1$ wires in their reference state that are placed parallel to each other. If such wires fluctuated independently, these would have produced a total of $F_3 \times F_3 = 4$ Fock states. There are two distinct ways of placing these wires with respect to each other such that they do not fluctuate independently and no inert

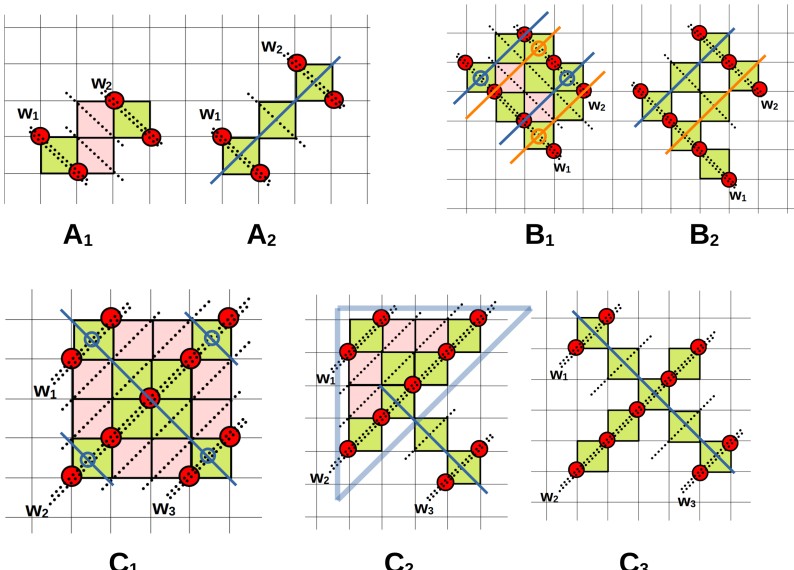

Figure 8: Examples of wire decomposition of quantum drums shown here. The drums labeled $A_1$ and $A_2$ can be constructed from two parallel wires both with $N_p = 1$, the drums labeled $B_1$ and $B_2$ can be constructed from two parallel wires with $N_p = 3$ and $N_p = 2$ while the drums labeled $C_1$, $C_2$, and $C_3$ can be constructed from three parallel wires, where two of them have $N_p = 1$ and one has $N_p = 4$. The wires in all the panels are indicated by double dashed lines and also labeled by $w_1$, $w_2$, $w_3$. The red filled dots in all panels represent up-spins (bosons). The perpendicular wires that can be generated from the Fock state shown for each drum are indicated by bold lines in blue and orange. Additionally, the plaquettes marked by open blue (orange) circles at their centers in $B_1$ and $C_1$ represent the locations of one-plaquette excitations that generate additional Fock states that cannot be represented by excitations of alternate parallel wires in these two cases. In the drum $C_2$, a smaller drum made of 10 edge-sharing plaquettes (indicated by thick blue region enclosing it) fluctuates simultaneously, with the wire $w_3$. The green (pink) plaquettes in the drums shown in all panels follow the same convention as used in Fig. 1.

up-spins (bosons) are created. These are shown as the quantum drums marked by $A_1$ and $A_2$ in Fig. 8. While the drum $A_1$ generates a Hilbert space fragment with 3 Fock states, the drum $A_2$ generates one with 5 Fock states. In the drum indicated by $A_1$ (Fig. 8), the wire $w_1$ ($w_2$) (indicated by double dotted lines in Fig. 8) can fluctuate to generate both its Fock states only if $w_2$ ($w_1$) is held fixed in its reference state. Thus, the two wires $w_1$ and $w_2$ *cannot* fluctuate simultaneously in $A_1$ and produce $2F_3 - 1 = 3$ states. On the other hand, in the drum $A_2$ (Fig. 8), both the wires $w_1$ and $w_2$ (indicated by double dotted lines in Fig. 8) *can* fluctuate simultaneously without producing a Fock state that violates the hard-core constraints. Additionally, performing a ring-exchange from the reference state on both the plaquettes that represent $w_1$ and $w_2$ generates the reference state for another wire with $N_p = 3$ that is *perpendicular* to $w_1$ and $w_2$ (shown as a blue line in the drum marked $A_2$ in Fig. 8). The Fock state obtained from a ring-exchange on the middle plaquette from the reference state of this $N_p = 3$ wire cannot be represented by combining any of the Fock states generated from the $w_1$ and $w_2$ wires and accounts for the total $F_3^2 + 1 = 5$ Fock states for the drum $A_2$. In the case of $A_1$ [$A_2$], the minimum distance between the parallel wires $w_1$ and $w_2$ equals $(3/2)\sqrt{2}$ [$2\sqrt{2}$] (Fig. 8).

The drums labeled $B_1$ and $B_2$ in Fig. 8 represent more complicated cases that arise when two parallel wires (of unequal lengths) in their reference states, one with $N_p = 3$ and another with $N_p = 2$, are brought close to each other such that the minimum distance between the wires equal $(3/2)\sqrt{2}$ and $2\sqrt{2}$ respectively. If these two wires fluctuated independently, these would have generated $F_5 \times F_4 = 15$ Fock states. However, the drum $B_1$ generates a fragment with 11 Fock states while the drum $B_2$ generates a fragment with 18 Fock states. In the drum $B_1$ (Fig. 8), the wire $w_1$ ($w_2$), shown by double dotted lines in Fig. 8, can fluctuate to generate all its Fock states only if the other wire $w_2$ ($w_1$) is held fixed in its reference state. Such wire fluctuations lead to $F_5 + F_4 - 1 = 7$ states. Two cases where fluctuations of a perpendicular wire (indicated by the top blue line and the bottom orange line respectively in the drum marked $B_1$ in Fig. 8) when the other wire parallel to it at a distance $(3/2)\sqrt{2}$ (indicated by the bottom blue line and the top orange line respectively in Fig. 8) is kept fixed in its reference state generates an additional 2 Fock states. The remaining 2 Fock states in the fragment are generated by two separate cases of ring-exchanges on two plaquettes together [indicated by the plaquettes with an open circle of the same color (blue and orange) at their centers] that are separated by 3 lattice units along $x/y$ as shown in Fig. 8 (panel marked $B_1$). On the other hand, in the drum $B_2$ (Fig. 8), both the wires $w_1$ and $w_2$ (indicated by double dotted lines in Fig. 8) can fluctuate simultaneously to generate all their Fock states without violating the hard-core constraints. Fluctuations of $w_1$ and $w_2$ in the drum $B_2$ cannot, however, generate any Fock state with two up-spins (bosons) along the diagonal parallel to $w_1$, $w_2$ on any of the two plaquettes that are not part of $w_1$ and $w_2$. The extra $18 - (F_5 \times F_4) = 3$ Fock states are generated from ring-exchange moves in any one of these two plaquettes starting with Fock states obtained from the fluctuations of $w_1$ and $w_2$ that can be represented as the reference state of a $N_p = 3$ wire perpendicular to both $w_1$ and $w_2$ (shown by a blue and an orange line perpendicular to $w_1$, $w_2$ in Fig. 8) and containing one of these two plaquettes.

Finally, we consider a case where three parallel wires in their reference states, $w_1$ with $N_p = 1$, $w_2$ with $N_p = 4$, and $w_3$ with $N_p = 1$, are brought close to each other to generate three different drums labeled $C_1$, $C_2$, and $C_3$ in Fig. 8. While independent fluctuations of these three wires generate $F_3 \times F_6 \times F_3 = 32$ Fock states, the fragment generated by the drum $C_1$ contains 24 Fock states, by the drum $C_2$ contains 28 Fock states, and by the drum $C_3$ contains 42 Fock states, respectively. In the drum $C_1$ (Fig. 8), the wire $w_1$ ($w_3$) can only access all its Fock states if $w_2$ is held fixed in its reference state (with these wires indicated by double dotted lines in the drum $C_1$ in Fig. 8). Similarly, the wire $w_2$ can only access all its Fock states if both $w_1$ and $w_3$ are fixed to their reference states in $C_1$. This generates a total $F_3^2 + F_6 - 1 = 11$ Fock states. An additional 5 Fock states of drum $C_1$ are generated by similar wire fluctuations of parallel wires separated by $(3/2)\sqrt{2}$ but perpendicular to $w_1$, $w_2$, $w_3$ (indicated by blue lines in drum $C_1$ in Fig. 8). Finally, the remaining 8 Fock states in $C_1$ are generated by simultaneous ring-exchanges on two/three of the four corner plaquettes (marked by blue circles at the centres of the corresponding plaquettes in $C_1$ in Fig. 8) that are separated from each other/from a corner plaquette by 3 lattice units in the $x$ or $y$ direction. On the other hand, in the drum $C_3$ (Fig. 8), all the wires, $w_1$, $w_2$ and $w_3$ (indicated by double dotted lines in Fig. 8), can fluctuate simultaneously without violating the hard-core constraints. Furthermore, fluctuations in $w_1$, $w_2$ and $w_3$ generates a new open channel of fluctuations in the form of a wire with $N_p = 5$ plaquettes in the direction perpendicular to these wires (indicated by a blue line in $C_3$ in Fig. 8) which generates an additional 10 Fock states besides the $F_3 \times F_6 \times F_3 = 32$ Fock states generated from $w_1$, $w_2$, $w_3$. The drum marked as $C_2$ in Fig. 8 represents an interesting intermediate case between $C_1$ and $C_3$ where the parallel wires $w_1$ and $w_2$ ($w_2$ and $w_3$) are at a distance $(3/2)\sqrt{2}$ ($2\sqrt{2}$) from each other. The smaller drum containing wires $w_1$, $w_2$ and composed of 10 edge-sharing plaquettes (marked by the blue region in the drum $C_2$ in Fig. 8) can fluctuate simultaneously with the wire $w_3$. This leads to a total $13 \times 2 = 26$ Fock states (see Fig. 18

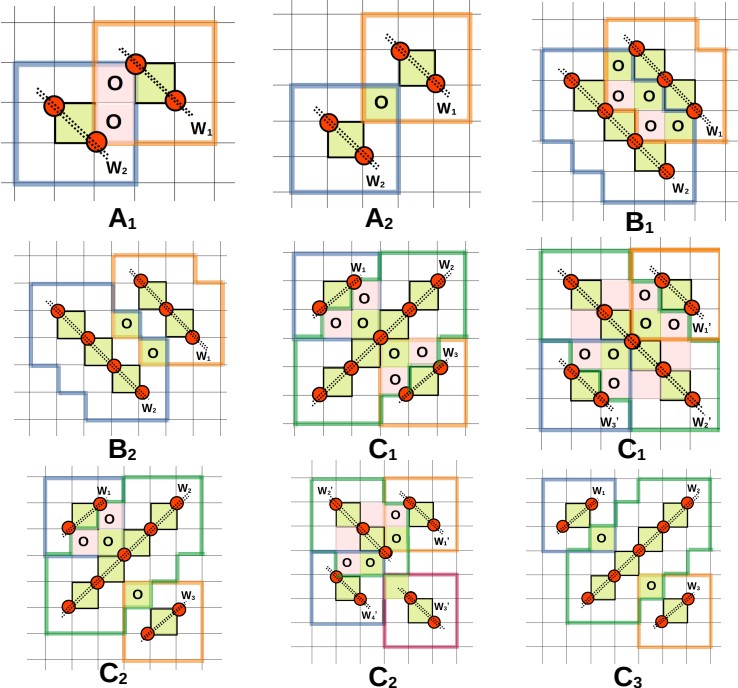

Figure 9: The drums $A_1$ to $C_3$ shown in Fig. 8 can be constructed using the overlap of the shielding regions of parallel wires as shown. The wires are indicated by double dashed lines in each panel. The red filled dots in all panels represent up-spins (bosons). The boundaries of the shielding region of each wire is shown using thick lines of different colors and the plaquettes formed by the overlap of the shielding regions are indicated by "o". The green (pink) plaquettes in the drums shown in all panels follow the same convention as used in Fig. 1.

for the 13 Fock states of the smaller drum made by the wires $w_1$ and $w_2$). The additional states are generated from an extra open channel for fluctuations along a $N_p = 3$ wire perpendicular to $w_1, w_2, w_3$ containing the $w_3$ plaquette as its right-most plaquette as indicated by the blue line perpendicular to $w_1, w_2, w_3$ in the drum $C_2$ in Fig. 8. This results in 2 Fock states where a ring-move is performed on the reference state of this $N_p = 3$ wire using the plaquette excluded from both the smaller drum composed of 10 edge-sharing plaquettes and $w_3$.

These examples demonstrate the general principle that given $n$ parallel wires (with unequal lengths in general) in their reference state, it is optimal to place these such that all the wires can fluctuate simultaneously and that these fluctuations additionally generate the maximum number of longest-possible wires perpendicular to the original wires as extra open channels of fluctuations to maximize the fragment size generated by the resulting drum. Both these conditions are satisfied by appropriate drums composed of only vertex-sharing plaquettes as shown in Fig. 8 (panels marked $A_2$, $B_2$ and $C_3$).

### 2.4.2 Constructing entire drums from wire-decomposed reference states

The wire-decomposed reference states introduced here provide an efficient route to construct the entire quantum drums associated with them. For this, we note that the shielding region of a single wire consists of all external plaquettes that share an edge or a vertex with the plaquettes that belong to the wire. The different wires in a wire-decomposed reference state are coupled to each other because (i) shielding regions of different parallel wires with spacing $(3/2)\sqrt{2}$ or $2\sqrt{2}$ have overlapping plaquettes and/or (ii) shielding regions of different parallel wires



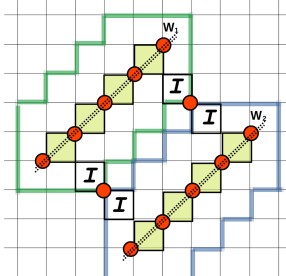 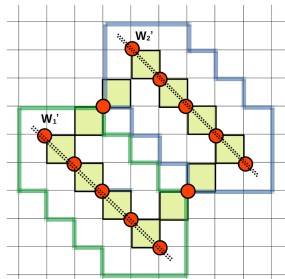

Figure 10: Two parallel wires $w_1$ and $w_2$ (see left panel) and two other parallel wires $w_1'$ and $w_2'$ (see right panel) indicated by double dotted lines, both in their reference state and both with $N_p = 4$, seperated by a distance of $3\sqrt{2}$. The wires $w_1$, $w_2$ are perpendicular to the wires $w_1'$, $w_2'$. In both panels, two up-spins (bosons) are located on two common sites between the boundaries of the shielding regions, shown using thick lines, of the wires. The red filled circles in both panels indicate up-spins. The green plaquettes in the drums shown in both panels follow the same convention as used in Fig. 1, with the right panel showing all the plaquettes that belong to this drum. In the left panel, plaquettes of the shielding regions of $w_1$ and $w_2$ that contain unpaired up-spins (bosons) are indicated by "I".

seperated by $3\sqrt{2}$ are coupled by unpaired up-spins (bosons). The overlapping plaquettes between different shielding regions (for (i)) and the plaquettes of the shielding region(s) that also contain any unpaired up-spin (boson) (for (ii)) provide the *remaining* plaquettes of the quantum drums starting from wire-decomposed Fock states.

Fig. 9 illustrates the procedure for all the quantum drums shown in Fig. 8. For the drums $A_1$, $A_2$, $B_1$, $B_2$ and $C_3$, it is sufficient to consider the overlap of the shielding regions of the parallel wires shown as double dashed lines in Fig. 8 as can be seen from Fig. 9, where the overlapping plaquettes of the shielding regions have been denoted by "o" in all panels of Fig. 9. The drums $C_1$ and $C_2$ present more interesting cases where this construction only identifies a subset of plaquettes that belong to the corresponding drum (Fig. 9). However, starting from the reference wire-decomposed state of the original parallel wires, it is easy to perform ring-exchange moves on a subset of the plaquettes that belong to these wires to create another wire-decomposed Fock state that can be viewed as parallel wires in their reference state, but in the perpendicular direction to the original wires (Fig. 9). The overlap of the shielding regions of these new wires gives the remaining plaquettes that are part of the quantum drum for both $C_1$ and $C_2$ (Fig. 9).

In Fig. 10, we show an example of a wire-decomposed Fock state where two parallel wires are separated by a distance of $3\sqrt{2}$. While the shielding regions of such wires do not have any overlapping plaquettes due to the increased distance between the parallel wires, these wires can still be coupled to each other to make a larger drum by placing unpaired up-spins (bosons) at a distance of $(3/2)\sqrt{2}$ from both wires in a subset of the common sites between the boundaries of the shielding regions of both the wires. In Fig. 10, two parallel wires $w_1$ and $w_2$, both with $N_p = 4$ and in their reference state, are placed at a distance of $3\sqrt{2}$ from each other (Fig. 10, left panel). Two unpaired up-spins (bosons) are placed on two of the common sites of the boundaries of the shielding regions of both the wires (the boundaries of the shielding regions are shown using thick green (blue) lines for $w_1$ ($w_2$) in Fig. 10, left panel). The plaquettes of the shielding regions that contain the unpaired up-spins (denoted by "I") then provide the remaining plaquettes of the entire quantum drum associated with this wire-decomposed reference state. Note that the wires $w_1$ and $w_2$ can fluctuate simultaneously

to generate all their Fock states in spite of the up-spins (bosons) on the boundaries of the shielding regions. Performing appropriate ring-exchanges on this reference state (Fig. 10, left panel), it is easy to get a Fock state that can be viewed as two parallel wires, $w_1'$ and $w_2'$, that are both perpendicular to $w_1$, $w_2$ and again separated by $3\sqrt{2}$ with two up-spins (bosons) on two of the common sites shared by the boundaries of the shielding regions of $w_1'$ and $w_2'$. This shows that four other plaquettes (apart from the ones shaded in Fig. 10, left panel) are part of the bigger drum (see Fig. 10, right panel) and that all the Fock states can be generated from simultaneous fluctuations of either $w_1$, $w_2$ or $w_1'$, $w_2'$, again generating a quantum drum with only vertex-sharing plaquettes.

## 3 Strong Hilbert space fragmentation

In this section, we show strong Hilbert space fragmentation for this kinematically constrained 2D model (Eq. 2) defined on a $L_x \times L_y$ rectangular lattice with OBC as $L_x, L_y \gg 1$. We first discuss numerical evidence from ED in Sec. 3.1. The wire decomposition of quantum drums introduced earlier is used in Sec. 3.2 to calculate the scaling of the Hilbert space fragment for drums composed of two long parallel wires. The wire decomposition is further used to identify the nature of the quantum drums that define the largest Krylov subspaces as a function of the density of up-spins (bosons) in Sec. 3.3. We then invoke the standard typicality argument [78] and show that typical initial states that belong to these large Krylov subspaces violate thermalization with respect to the full Hilbert space.

### 3.1 Numerical evidence of strong fragmentation from exact diagonalization

One procedure to distinguish between weak and strong fragmentation [52,53] involves monitering the ratio of the largest Hilbert space fragment (denoted by $\max[\mathcal{D}_{f,n}]$) to the total Hilbert space dimension (denoted by $\mathcal{D}_n$) in a sector with a *fixed* density (denoted by $n$) of up-spins (bosons) for different system sizes. We stress here that only the global symmetry of total magnetization conservation and its associated density is relevant for this analysis since internal symmetries like reflections etc can always be removed by adding suitable diagonal terms to $H$ in the computational basis that do not connect the different Hilbert space fragments. If the ratio $\max[\mathcal{D}_{f,n}]/\mathcal{D}_n$ behaves as $\exp(-\gamma N)$ with $\gamma > 0$ as the number of sites in the system, $N$, diverges, it implies strong fragmentation; in contrast, if it approaches 1 as $N \gg 1$, it implies weak fragmentation.

Using exact enumeration techniques, we calculate the Hilbert space dimension, $\mathcal{D}$, for a fixed number of up-spins (bosons), $N_b$, for a variety of rectangular lattices of dimension $L_x \times L_y$ with OBC (see Fig. 11, left panel) which shows that $\mathcal{D}$ is maximized when $n = N_b/(L_x L_y) = 1/4$. We then focus on this particular density of up-spins (bosons) $n = 1/4$ as well as two other densities $n = 3/10$ and $n = 1/3$ to show the scaling of $\max[\mathcal{D}_{f,n}]/\mathcal{D}_n$ for fixed $n$ as a function of $N = L_x L_y$ in Fig. 11, right panel using data from exact enumeration. The data for these limited system sizes already clearly indicate that $\max[\mathcal{D}_{f,n}]/\mathcal{D}_n \sim \exp(-\gamma N)$ with $\gamma$ depending on the density of up-spins (bosons), $n$, and thus points towards strong Hilbert space fragmentation in this 2D model. In Sec. 3.3, we will show that the dimension of the largest Hilbert space fragment generated in this model scales as $(\varphi^{1/4})^{L_x L_y}$ for $L_x, L_y \gg 1$ when $n = 1/4$, where $\varphi = (1 + \sqrt{5})/2$ (golden ratio). Given that the total Hilbert space dimension scales as $\kappa^{L_x L_y}$ for $L_x, L_y \gg 1$ [76], it is reassuring to see that $(\varphi^{1/4}/\kappa)^{L_x L_y}$ (dotted curve in Fig. 11, right panel) closely follows the data for $\max[\mathcal{D}_{f,n}]/\mathcal{D}_n$ at $n = 1/4$ (Fig. 11, right panel) since the density $n = 1/4$ gives the dominant contribution to the total Hilbert space (Fig. 11, left panel) at these system sizes.

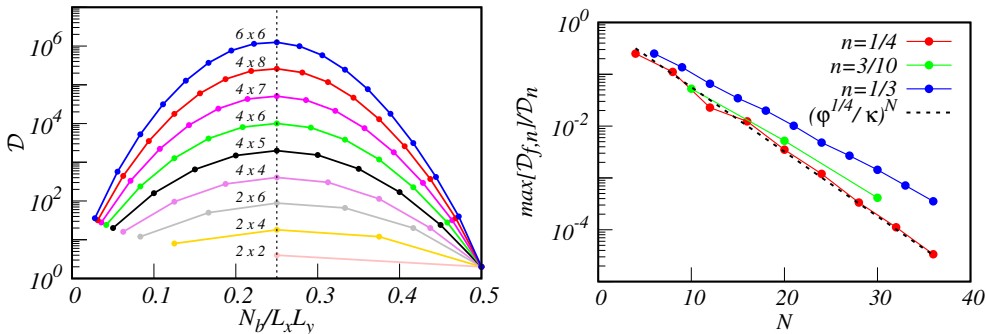

Figure 11: (Left panel) The total Hilbert space dimension $\mathcal{D}$ as a function of the number of up-spins, $N_b$, for rectangular lattices $L_x \times L_y$ of various dimensions. The thin dotted vertical line is at $n = N_b/(L_x L_y) = 1/4$. (Right panel) The behavior of the ratio of the dimension of the largest Hilbert space fragment and the total Hilbert space dimension for the magnetization sector for up-spin densities $n = 1/4$ (blue), $n = 3/10$ (green) and $n = 1/3$ (red) shown for rectangular lattices with dimension $L_x \times L_y$ and OBC as a function of the system size $N = L_x L_y$. The dotted curve shows the function $(\varphi^{1/4}/\kappa)^N$ where $\varphi = (1+\sqrt{5})/2$ (golden ratio) and $\kappa \approx 1.503 \cdots$ (hard square entropy constant).

## 3.2 Quantum drums generated from two long parallel wires

While Sec. 2.4 illustrated the wire decomposition using small drums, this concept becomes most efficient to calculate the scaling of the resulting fragment size when macroscopic drums are considered. Since the wire-decomposed reference Fock states of such drums typically contain long parallel wires, we will consider the warm-up exercise of wire-decomposed Fock states with two such wires in this section. More precisely, we consider two parallel wires, each with $N_p$ plaquettes such that $N_p \gg 1$, where both the wires are in their reference Fock states. Bringing these wires (denoted by $w_1$ and $w_2$ in top-left and bottom-left panels of Fig. 12) at a distance of $(3/2)\sqrt{2}$ $[2\sqrt{2}]$ from each other generates a drum with only edge-sharing [vertex-sharing] plaquettes as shown in panel A [panel B] of Fig. 12. Putting two such parallel wires closer than $(3/2)\sqrt{2}$ (further than $2\sqrt{2}$) leads to both the wires being inert (independent of each other). As we will show below, the number of Fock states in the corresponding drum scales as $\varphi^{N_p}$ $[\varphi^{2N_p}]$, where $\varphi = (1+\sqrt{5})/2$ denotes the golden ratio, for $N_p \gg 1$ when the two parallel wires, $w_1$ and $w_2$, are at a distance of $(3/2)\sqrt{2}$ (Fig. 12, panel A) $[2\sqrt{2}]$ (Fig. 12, panel B) from each other.

Focusing on the case where $w_1$ and $w_2$ are in their reference Fock states and separated from each other by a distance $(3/2)\sqrt{2}$ (Fig. 12, top left panel), the wire $w_1$ $[w_2]$ can access all its allowed Fock states if and only if $w_2$ $[w_1]$ is kept fixed in its reference state. This immediately shows that $2\varphi^{N_p} < \mathcal{N}_A$ for $N_p \gg 1$, where $\mathcal{N}_A$ denotes the total number of Fock states for the quantum drum shown in panel A of Fig. 12. Crucially, not all simultaneous fluctuations of $w_1$ and $w_2$ are disallowed. E.g., starting from the Fock state shown in Fig. 12 (top left), independent fluctuations of elementary plaquettes that are arranged in a regular pattern generated by the primitive vectors $3\hat{x}$ and $3\hat{y}$, as indicated by open blue dots at the centers of such plaquettes in Fig. 12 (top left), are allowed. The number of Fock states generated from such independent one-plaquette excitations scale as $3(2^{N_p/3})(2^{N_p/3})$ where the factor of 3 is due to the inequivalent arrangements of this regular pattern. Moreover, the Fock state shown in Fig. 12 (top right) can be generated using a sequence of ring-exchange moves from the reference Fock state shown in Fig. 12 (top left), which again allows for independent one-plaquette

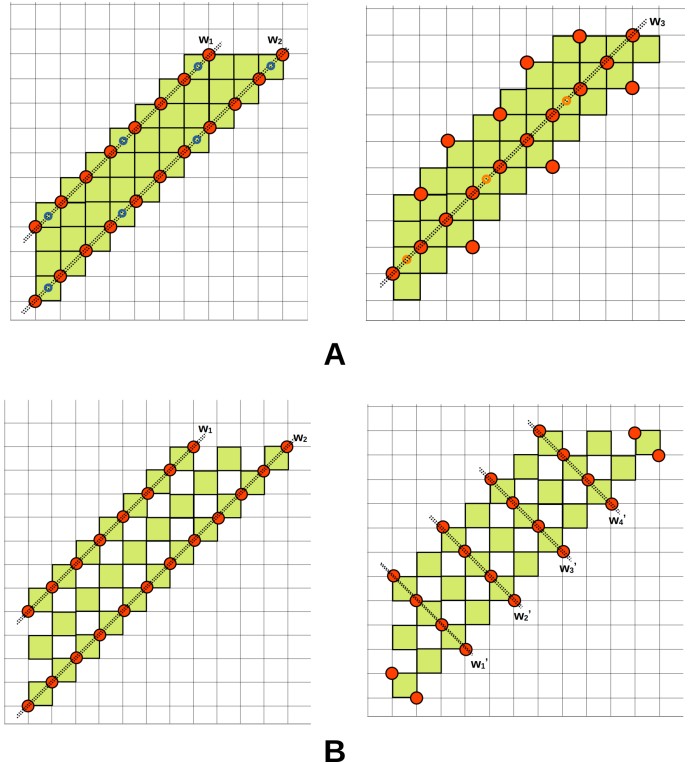

**Figure 12:** A section of two quantum drums that are generated by placing two long parallel wires, denoted by $w_1$ and $w_2$, in their reference states at a mutual separation of $(3/2)\sqrt{2}$ (panel A) and $2\sqrt{2}$ (panel B) respectively, shown here . The green plaquettes in the quantum drums follow the same convention as used in Fig. 1. The filled red dots indicate up-spins (bosons) in all the panels and represent just one of the many possible Fock states of the corresponding drum. The elementary plaquettes indicated by open blue circles (open orange circles) in the top-left figure (top-right figure) in panel A allow for independent ring-exchange moves starting from the Fock state shown in the same figure. Note that only these plaquettes allow for ring-exchange moves for the wire $w_3$ in the top-right figure of panel A. In panel B, the Fock states can be generated either from the simultaneous fluctuations of the wires, $w_1$ and $w_2$, as shown in bottom-left figure (panel B) or from the simultaneous fluctuations of parallel short wires (four of them, $w_1'$, $w_2'$, $w_3'$, $w_4'$ shown in the bottom-right figure (panel B)) that are perpendicular to $w_1$ and $w_2$.

excitations along the wire $w_3$, where such plaquettes are again arranged in a regular pattern generated by the primitive vectors $3\hat{x}$ and $3\hat{y}$, as indicated by open orange dots at the centers of such plaquettes in Fig. 12 (top right). The number of such Fock states scale as $6(2^{N_p/3})$ where the factor of $3+3$ is due to the 3 inequivalent arrangements of such a regular pattern along $w_3$ and 2 inequivalent ways of placing $w_3$.

A combination of all these independent one-plaquette excitations and excitations of $w_1$ ($w_2$) keeping $w_2$ ($w_1$) fixed in its reference Fock state generates all the Fock states of the quantum drum shown in panel A of Fig. 12 with certain Fock states being produced multiple times. Thus, in the limit $N_p \gg 1$, we get that

$$2(\varphi)^{N_p} < \mathcal{N}_A < 2(\varphi)^{N_p} + 3(2^{2/3})^{N_p} + 6(2^{1/3})^{N_p} \Rightarrow \mathcal{N}_A \sim (\varphi)^{N_p}, \tag{13}$$

where the final result in Eq. 13 follows since $\varphi > 2^{2/3} > 2^{1/3}$.

We now consider the case of the quantum drum shown in Fig. 12 (bottom left) which is generated from two parallel wires, $w_1$ and $w_2$, both in their reference Fock states such that these wires are seperated by a distance of $2\sqrt{2}$. Since simultaneous fluctuations of both $w_1$ and $w_2$ are allowed in this case without violating any hard-core constraints, the total number of Fock states, $\mathcal{N}_B$, in the quantum drum shown in panel B of Fig. 12 is bounded below by $(\varphi^2)^{N_p} < \mathcal{N}_B$. The remaining Fock states are generated from a subset of the Fock states generated from the simultaneous fluctuations of parallel wires that are perpendicular to the original $w_1$ and $w_2$ and separated by $2\sqrt{2}$ lattice units from each other (e.g., a subset of such wires are marked as $w_1'$, $w_2'$, $w_3'$ and $w_4'$ in the bottom right panel of Fig. 12). Since each of these short wires are composed of 3 plaquettes (Fig. 12, bottom right), it follows that each such wire contributes $F_5$ states with there being $N_p/2$ of them which fluctuate independently. The resulting number of Fock states equal $2(F_5)^{N_p/2}$ where the factor of 2 is due to the 2 inequivalent arrangements of such short parallel wires. Thus, in the limit $N_p \gg 1$, we get that

$$(\varphi^2)^{N_p} < \mathcal{N}_B < (\varphi^2)^{N_p} + 2(\sqrt{F_5})^{N_p} \Rightarrow \mathcal{N}_B \sim (\varphi^2)^{N_p}, \tag{14}$$

where the final result in Eq. 14 follows since $\varphi^2 > \sqrt{F_5}$.

Eq. 13 and Eq. 14 show that an important simplification emerges when macroscopically long parallel wires are involved in the construction of the reference Fock state of a drum. The correct scaling of the Hilbert space fragment dimension in both drums is obtained simply from the fluctuations of those long wires that can access *all* their states simultaneously while the rest of the fluctuations can be considered as subdominant. When the wires $w_1$ and $w_2$ are at a distance of $(3/2)\sqrt{2}$, $w_1$ ($w_2$) can access all its states only when $w_2$ ($w_1$) is fixed to its reference Fock state implying that $\mathcal{N}_A \sim (\varphi)^{N_p} + (\varphi)^{N_p} \sim (\varphi)^{N_p}$. When $w_1$ and $w_2$ are at a distance of $2\sqrt{2}$ from each other, both wires can access all their states simultaneously to give $\mathcal{N}_B \sim (\varphi)^{N_p} \cdot (\varphi)^{N_p} \sim (\varphi^2)^{N_p}$.

### 3.3 Large Krylov subspaces and absence of ETH-predicted thermalization

In contrast to systems that display weak Hilbert space fragmentation, typical initial states in strongly fragmented systems *do not* thermalize with respect to the full Hilbert space due to the absence of a single dominant Krylov subspace in the thermodynamic limit. We now consider the fate of typical unentangled initial states for a large system, say a $L \times L$ lattice with OBC where $L \gg 1$, under unitary time evolution with $H$. Given the $E$ to $-E$ symmetry of the many-body spectrum, typical initial states at any fixed density of up-spins (bosons), $n$, will have an average energy per site equal to $\langle E \rangle / L^2 = 0$ for $L \gg 1$. Thermalization in the full Hilbert space (ETH) with fixed $n$ will imply that such an initial state with a macroscopic number of up-spins (bosons) should thermalize to the infinite temperature ensemble (ITE) with $n$ fixed by the initial condition as far as local operators are concerned. Thus, ETH-predicted thermalization implies that local operators lose *all* memory of the initial state, except its conserved $n > 0$, under unitary evolution with $H$.

Given a typical initial state with an extensive number of up-spins (bosons), it can be categorized in one of the following five classes:

1. The initial state is an inert Fock state which forms a 1-dimensional fragment on its own.

2. The initial state is consistent with a finite number of finite-sized drums when $L \gg 1$.

3. The initial state is consistent with an extensive number of finite-sized drums when $L \gg 1$.

4. The initial state is consistent with the presence of one or more (subextensive) quasi-1D drums with a typical linear dimension of $O(L)$ as $L \gg 1$.

5. The initial state is consistent with the presence of one or more 2D drums with a typical linear dimension of $O(L)$ as $L \gg 1$.

Initial states in class 1 and 2 clearly belong to Krylov subspaces that remain of size $O(l)$, where $l$ stays finite even in the thermodynamic limit and cannot thermalize with respect to the full Hilbert space. Initial states in class 3, 4, 5 belong to large Krylov spaces whose size scale exponentially with $L^2$ when $L \gg 1$. Thus, it is not immediately obvious whether such states evade ETH-predicted thermalization with respect to the full Hilbert space or not.

Initial states in class 3 contain an extensive number of inert down-spins when all the boundary sites of the shielding regions of the drums are considered together. For initial states in class 4, one simply needs to consider local operators that have support from sites on opposite sides of a quasi-1D drum of linear dimension $O(L)$. Such local operators evade ETH-predicted thermalization with respect to the full Hilbert space since all the sites that compose such a local operator cannot be part of the same quantum drum and are, therefore, dynamically disconnected and retain memory of the initial state.

Initial states in class 5 are more subtle since almost all local operators contain sites in the interior of a single 2D quantum drum and it is not immediately clear which local operators evade ETH-predicted thermalization unlike initial states in class 3 and 4. We take such a 2D quantum drum to cover the entire $L \times L$ lattice without any loss of generality. Following Sec. 2.4, a reference wire-decomposed Fock state for such a 2D drum can be composed by bringing $O(L)$ parallel wires together, typically of length $O(L)$, in their reference states. Examples of some 2D drums are shown in Fig. 13 (three panels) and in Fig. 14 (right panel). When macroscopic 2D drums *can* be formed, their fragment dimensions dominate over those generated by a collection of an extensive number of finite-sized drums or a subextensive number of quasi-1D drums at the same density of up-spins (bosons) since new channels of fluctuations open up in these 2D drums. However, at low densities of up-spins (bosons), $n \ll 1$, it is clear that macroscopic 2D drums cannot be formed purely from geometric considerations and there must exist some critical $n_c$, only above which these 2D drums start dominating statistically.

In a wire-decomposed Fock state, the macroscopic parallel wires cannot be farther than a distance of $3\sqrt{2}$ from each other so that such wires can be at least coupled to each other using unpaired up-spins (bosons). One such example of a 2D drum is shown in Fig. 13 (panel A) at $n = 2/9$. While the parallel wires contain 3/4 of the up-spins (bosons), the unpaired up-spins (bosons) account for the rest of the 1/4 up-spins (bosons) contained in the 2D drum. Note that the density of the unpaired up-spins (bosons) can be reduced to an arbitrarily low number to still give a 2D, though highly anistropic, drum. This immediately gives $n_c = 1/6$. Thus, typical initial states for $n \in (0, 1/6)$ evade ETH-predicted thermalization simply from the presence of local operators that retain the memory of their initial condition by being a class 3 or class 4 state.

For $n > 1/6$, it becomes important to be able to calculate the scaling of the fragment size of a macroscopic 2D drum given its wire-decomposed reference state. Given that the number of Fock states accessible to a single wire of length $l$ equals $F_{l+2}$ (Eq. 29) where $F_n$ are the Fibonacci numbers, the fragment size scales exponentially with increasing $l$ as $\varphi^l$ for $l \gg 1$, where $\varphi = (1 + \sqrt{5})/2 \approx 1.618 \cdots$ is the golden ratio. Thus, if wire-decomposed reference states of 2D drum consist of parallel wires that can fluctuate simultaneously to access all their internal states, then the corresponding number of Fock states generated equals

$$\varphi^{(L_1 + L_2 + L_3 + L_4 + \cdots)}, \tag{15}$$

where $L_1, L_2, L_3, L_4$ etc denote the lengths of these wires, which are typically $O(L)$ for macroscopic drums. Eq. 15 already shows that the corresponding Hilbert space fragment grows exponentially with the system size. Just like the case of quantum drums formed out of two long parallel wires as discussed in Sec. 3.2, the scaling of the total number of Fock states in

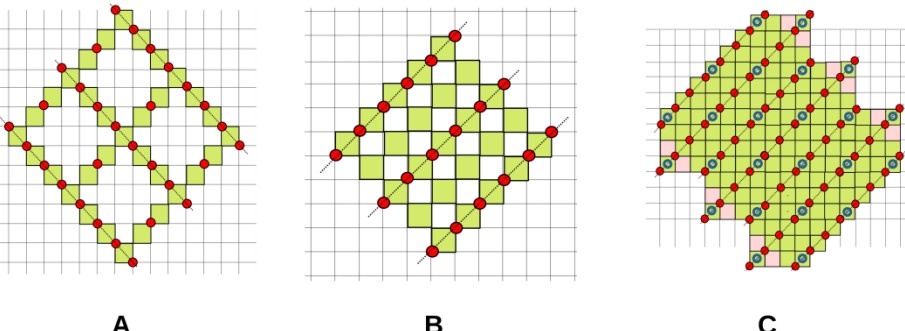

**A**             **B**             **C**

Figure 13: Three examples of 2D drums shown here. The green (pink) plaquettes in all the quantum drums follow the same convention as used in Fig. 1. The filled red dots indicate up-spins in all the panels and represent just one of the many possible Fock states of the corresponding drum. The wire decomposition of each drum is indicated by thin dotted lines in each of the three panels. Panels A and B are composed of parallel wires that can fluctuate simultaneously and hence contain only vertex-sharing plaquettes. The checkerboard drum in panel B represents the maximum packing of such wires that gives the density of up-spins (bosons) to be 1/4. The close packed drum in panel C represents the maximum packing of wires such that none of them are inert that gives the density of up-spins (bosons) to be 1/3. The plaquettes where ring-exchanges can be simultaneously carried out are indicated by open blue circles at their centres in panel C.

2D drums composed of a macroscopic number of long parallel wires can be simply estimated by considering the number of ways in which such wires can fluctuate simultaneously as given in Eq. 15. The *missing Fock states* that cannot be accounted for by the wire fluctuations represented in Eq. 15 only give a subdominant correction for macroscopic drums. We will show this explictly by considering two important cases of such 2D quantum drums below.

We first consider a "checkerboard" drum (see panel B in Fig. 13) which represents the closest packing of parallel wires (indicated by dotted lines in panel B of Fig. 13) in their reference state such that all the wires can fluctuate simultaneously to access all their states. Comparing the representative Fock state of this drum shown in Fig. 13 (panel B) to the inert state with the maximum density of up-spins (bosons) that equals $n = 1/2$ (Fig. 14), we see that the former may be obtained from the latter by removing the up-spins (bosons) on alternate parallel wires from the inert state. This fixes the density of up-spins (bosons) to be $n = (1/2) \times (1/2) = 1/4$ for the checkerboard drum. All the Fock states of this drum can be generated (in fact, overcounted) by considering simultaneous fluctuations of wires along either of the diagonal directions of the square lattice with mutual separation of $2\sqrt{2}$ and also their shifted counterparts with a shift of $\sqrt{2}$ perpendicular to the direction of the wires. This immediately establishes that

$$(\varphi^{1/4})^{L^2} < \mathcal{N}_{\text{HSD,ch}} < 4(\varphi^{1/4})^{L^2}, \tag{16}$$

where $\mathcal{N}_{\text{HSD,ch}}$ equals the number of Fock states for this drum when $L \gg 1$. Thus, we get that

$$\mathcal{N}_{\text{HSD,ch}} \sim (\varphi^{1/4})^{L^2} \approx (1.12784\cdots)^{L^2}, \tag{17}$$

for the 2D checkerboard drum that accomodates the maximum density of simultaneously fluctuating parallel wires, resulting in a density of up-spins (bosons) that we denote as $n_{\text{ch}} = 1/4$ henceforth.

As shown in Sec. 2.4, the closest distance of approach between two parallel wires in their reference state equals $(3/2)\sqrt{2}$ such that these do not become inert. Extending this to 2D,

one gets a "close packed drum" as shown in Fig. 13 (panel C) where the parallel wires are indicated by dotted lines. Unlike the checkerboard drum, this 2D drum is composed of only edge-sharing plaquettes and its interior has no unshaded plaquettes that do not belong to the drum. The density of up-spins (bosons) for this close packed drum equals $n = 1/3$ which can be seen by comparing the reference Fock state shown in panel C of Fig. 13 to the inert state with the maximum density of up-spins (bosons), $n = 1/2$, (Fig. 14). We see that the former may be obtained from the latter by deleting the up-spins (bosons) on every two parallel wires and keeping every third wire intact from the inert state in a $1-0-0$ pattern. This gives one set of simultaneously flippable wires of the close packed drum, implying that $n = (1/6 + 1/6) = 1/3$ since two sets of such wires are needed to make up the close packed drum. As discussed in Sec. 2.4, all Fock states of such structures where the consecutive parallel wires are at a distance of $(3/2)\sqrt{2}$ can be generated from two types of excitations: (a) simultaneous fluctuations of every alternate parallel wire and (b) excitations of elementary plaquettes that are separated by 3 lattice units along $x$ or $y$ and thus simultaneously flippable. One set of such parallel wires (indicated by parallel dotted lines) and elementary plaquettes (indicated by open blue circles in the centers of the corresponding plaquettes) are shown in Fig. 13 (panel C). The scaling of the number of Fock states associated with the wire fluctuations can be simply calculated using Eq. 15 and gives $(\varphi^{1/6})^{L^2}$. The number of states generated from the simultaneously flippable elementary plaquettes can be calculated by simply noting that the result is identical to the one discussed in Sec. 2.3 since the elementary plaquettes have the same spatial arrangement in Fig. 7 as the marked plaquettes in Fig. 13 (panel C) thus giving the number of such excitations as $(2^{1/9})^{L^2}$ (Eq. 9). Furthermore, all the Fock states can be generated (in fact, overcounted) by considering all combinations of such parallel wires as well as their perpendicular counterparts and the simultaneously flippable elementary plaquettes and their lattice translations. This gives that

$$(\varphi^{1/6})^{L^2} < \mathcal{N}_{\text{HSD,cp}} < 4\left[(\varphi^{1/6})^{L^2} + (2^{1/9})^{L^2}\right],\tag{18}$$

where $\mathcal{N}_{\text{HSD,cp}}$ equals the fragment dimension for this drum when $L \gg 1$. Importantly, since $2^{1/9}/\varphi^{1/6} \approx 0.996819\cdots$, the above equation can be simplified to give

$$\mathcal{N}_{\text{HSD,cp}} \sim (\varphi^{1/6})^{L^2} \approx (1.08351\cdots)^{L^2},\tag{19}$$

for the close packed drum that accomodates the maximum density of non-inert parallel wires, resulting in a density of up-spins (bosons) that we denote as $n_{\text{cp}} = 1/3$ henceforth.

Now that we have established that Eq. 15 gives the correct scaling of Hilbert space fragment dimension for 2D quantum drums using these two examples, a straightforward approach to maximize the number of states produced in a fragment, given a certain density of up-spins (bosons), $n > 1/6$, is to consider 2D drums where *all* the parallel wires that compose the drums can fluctuate simultaneously. This automatically lead to drums made of only vertex-sharing plaquettes. Since such parallel wires can only have a minimum separation of $2\sqrt{2}$, this sets an upper bound on the density of up-spins (bosons) to be $n_{\text{ch}} = 1/4$ (the density for the checkerboard drum). Since the checkerboard drum maximizes Eq. 15, it generates the largest Hilbert space fragment for this model. For $n \in (1/6, 1/4]$, typical initial states belong to fragments generated by 2D drums composed of only vertex-sharing plaquettes since these dominate statistically.

An example of such a 2D drum at a lower density compared to $n = 1/4$ using unpaired up-spins (bosons) at a distance of $(3/2)\sqrt{2}$ to couple consecutive parallel wires that are $3\sqrt{2}$ distance apart is shown in panel A of Fig. 13. Comparing the reference Fock state shown in panel A of Fig. 13 to the inert state with the maximum density of up-spins (bosons), $n_b = 1/2$, (Fig. 14), we see that $n = (1/6) \times (1 + 1/3) = 2/9$ for this quantum drum. The parallel wires in panel A of Fig. 13 can be obtained by deleting every two wires in the inert state and

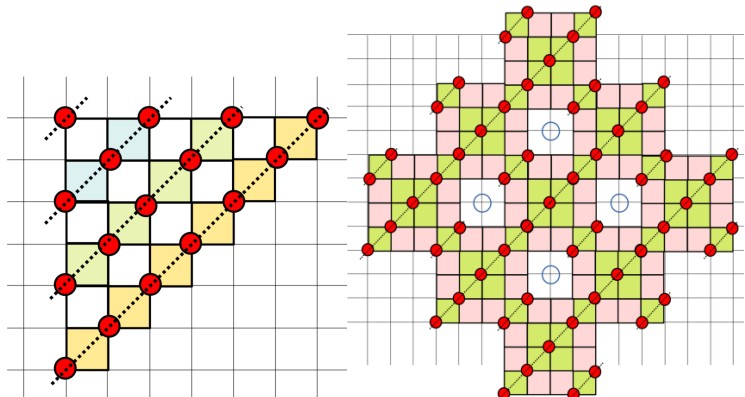

Figure 14: (Left panel) A section of the inert state with the maximum density of up-spins (bosons) equal to 1/2 shown with the filled red dots indicating up-spins. This state can be viewed in terms of parallel wires (indicated by dotted lines and shaded plaquettes of different colors) that are placed so close that they cannot fluctuate out of their reference states. (Right panel) Example of a 2D quantum drum with a density of up-spins (bosons) between 1/4 and 1/3. In this particular example, the density equals 5/18. The green (pink) plaquettes in this quantum drum follows the same convention as used in Fig. 1. The filled red dots indicate up-spins and represents just one of the many possible Fock states of the corresponding drum. The wire decomposition of the drum is indicated by thin dotted lines. The blue circles indicate inert down-spins that are not part of this quantum drum.

keeping every third wire intact in a $1-0-0$ pattern while the up-spins that do not belong to any wire in panel A of Fig. 13 can again be obtained by taking the same $1-0-0$ pattern of wires and deleting every two up-spins (bosons) and keeping every third up-spin (boson) in the surviving wires. The wire decomposition of the drum (shown in panel A, Fig. 13) shows that the fragment size scales as $(\varphi^{1/6})^{L^2} \approx (1.0835\cdots)^{L^2}$ which dominates over the fragment produced by closed-packed one-plaquette drums $(2^{1/9})^{L^2} \approx (1.08006\cdots)^{L^2}$ (see Sec. 2.3) at the same density $n = 2/9$.

Crucially, any 2D drum composed of vertex-sharing plaquettes alone contain an extensive number of unshaded plaquettes in its interior (Fig. 13, panels A and B) that do not belong to the drum. By definition, the two-spin local correlators $\langle(\sigma^z_{j_x,j_y} + 1)(\sigma^z_{j_x+1,j_y+1} + 1)\rangle$ and $\langle(\sigma^z_{j_x,j_y+1} + 1)(\sigma^z_{j_x+1,j_y} + 1)\rangle$ stay pinned to zero for any such unshaded plaquette during the time evolution induced by $H$ as two up-spins (bosons) cannot occupy the diagonals for these unshaded plaquettes. Thus, the intial states that belong to such 2D drums *retain* an extensive amount of local memory during time evolution with $H$ and *do not* satisfy ETH-pedicted thermalization for this model.

Furthermore, while the checkerboard drum (Fig. 13, panel B) does not contain any inert spin in its interior, drums made of vertex-sharing plaquettes with a lower density of up-spins (bosons) (e.g., Fig. 13, panel A) also contain an extensive number of sites in their interior that do not belong to the drum and thus harbor inert spins. It is useful to note here that if *all* unentangled initial states that satisfy the hard-core constraints in Eq. 2 are considered uniformly, then the average density of up-spins (bosons) equals $\langle n \rangle = 0.226570\cdots$ in the thermodynamic limit [76]. Thus, drums composed of only vertex-sharing plaquettes dominate statistically if typical initial states are considered *without putting any further restriction on n*.

2D drums with the largest fragment dimension for $n \in (1/4, 1/2]$ cannot be determined by just considering drums composed of vertex-sharing plaquettes. We will first show that typical

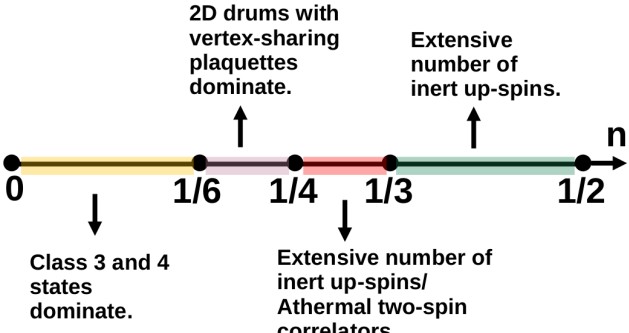

Figure 15: The fate of a typical initial state with a macroscopic number of up-spins (bosons), $n$, under unitary time evolution with $H$ is shown. Below $n_c = 1/6$, a typical initial state is either consistent with an extensive number of finite-sized drums (class 3) or a subextensive number of quasi-1D drums (class 4). For $n \in (1/6, 1/4)$, typical initial states belong to fragments produced by macroscopic 2D drums with vertex-sharing plaquettes. For $n \in (1/4, 1/3)$, a typical initial state contains either an extensive number of inert up-spins (bosons) or an extensive number of athermal next-nearest neighbor two-spin correlators. For $n > 1/3$, a typical initial state always contains an extensive number of inert up-spins. The largest Hilbert space fragment is produced by the 2D checkerboard drum (panel B of Fig. 13) at $n = 1/4$.

initial states that belong to large Krylov subspaces always contain an extensive number of inert up-spins (bosons) for $n \in (1/3, 1/2]$. As already discussed in this section, the close packing of long parallel wires such that neighboring wires can still fluctuate generates the close packed drum (Fig. 13, panel C) and a corresponding density of $n_{cp} = 1/3$. Adding an excess amount of up-spins (bosons) in the system such that $n = (1/3) + \delta$, where $\delta \in (0, 1/6)$, necessarily leads to an extensive number of inert wires locked in their reference state, with the density of such wires scaling as $\delta/6$. Thus, the number of inert up-spins (bosons) in typical initial states should scale as $\delta L^2/6$ for $n = (1/3) + \delta$ with $\delta \in (0, 1/6)$ when $n \in (1/4, 1/2]$. Thus, these states also evade ETH-predicted thermalization.

We now come to the nature of large Krylov subspaces where the density of up-spins (bosons) equals $n \in (1/4, 1/3)$ such that we can write $n = (1/4) + \gamma$ with $\gamma \in (0, 1/12)$. Three different possibilities exist here.

(i) One can start with the reference state of the checkerboard drum in Fig. 13 (panel B) and insert extra parallel wires in their reference state such that the distance between parallel wires equals $\sqrt{2}$ for a linear extent of $4\gamma L$ of the system while the rest of the system still has parallel wires that can fluctuate simultaneously. However, this immediately produces $O(4\gamma L^2)$ inert up-spins (bosons) in the system and thus such initial states evade thermalization. Since the size of the spatial region that harbors wires that can simultaneously fluctuate reduces from $L^2$ to $(1 - 4\gamma)L^2$, the fragment size scales as $\left(\varphi^{\frac{1}{4} - \gamma}\right)^{L^2}$ using Eq. 17 in this case.

(ii) One can have "phase-separated" drums, with the phase separation in one direction, such that different macroscopic regions are composed of close packed parallel wires in their reference state. One of these sets have parallel wires at a distance $(3/2)\sqrt{2}$ (with a local density $n = 1/3$) for a total linear extent of $(12\gamma)L$. The other regions comprise of parallel wires in their reference state at a distance $2\sqrt{2}$ (with a lower local density of $n = 1/4$) for the rest of the system to get the correct up-spin (boson) density. We refer the reader to the drum marked as $C_2$ in Fig. 8 for an illustration of this principle for a small drum where the parallel wires $w_1$ and $w_2$ ($w_2$ and $w_3$) are at a distance $(3/2)\sqrt{2}$ ($2\sqrt{2}$) with respect to each other. The leading scaling for the Hilbert space size of such drums can simply be obtained by considering inde-



pendent fluctuations of these "checkerboard drum" and "close packed drum" regions. Using Eq. 17 and Eq. 19, this immediately gives

$$\mathcal{N}_{\text{HSD,phase-sep}} \sim [\varphi^{1/6}]^{12\gamma L^2} [\varphi^{1/4}]^{(1-12\gamma)L^2} \sim \left(\varphi^{\frac{1}{4}-\gamma}\right)^{L^2}. \tag{20}$$

(iii) One can start with the close packed drum reference state and delete up-spins (bosons) in such a manner that another 2D drum with a lower density of up-spins (bosons) is created. We refer the reader to the 2D drum shown in the right panel of Fig. 14 where the reference Fock state of this particular drum can be produced from the reference Fock state of the close packed drum by deleting every third up-spin (boson) from alternate wires in their reference state. This gives a reduced density of up-spins (bosons) to be $(1/6)(1+2/3) = 5/18 \approx 0.2777\cdots$ as well as an extensive density of inert down-spins from sites that are not part of the drum. However, the scaling of the fragment size for such drums is upper-bounded by $\left(\varphi^{\frac{1}{6}}\right)^{L^2}$.

Fragments in (i) and (iii) dominate statistically for $n \in (1/4, 1/3)$ and typical initial states thus either contain an extensive number of inert spins or an extensive number $[O((1-12\gamma)L^2)]$ of unshaded plaquettes in the interior of "phase-separated" drums where two-spin local correlators $\langle(\sigma^z_{j_x,j_y} + 1)(\sigma^z_{j_x+1,j_y+1} + 1)\rangle$ and $\langle(\sigma^z_{j_x,j_y+1} + 1)(\sigma^z_{j_x+1,j_y} + 1)\rangle$ stay pinned to zero, thus evading ETH-predicted thermalization.

This completes our analysis for the lack of ETH-predicted thermalization in typical initial states for all $n \neq 1/3$. We see that typical initial states have either an extensive number of inert spins or an extensive number of two-spin next-nearest neighbor correlators that are pinned to athermal values or both. The situation is summarized in Fig. 15 as a function of the density of up-spins (bosons), $n$.

The case of the close packed drum (Fig. 13, panel C) with density of up-spins (bosons) $n = 1/3$ with a Hilbert space fragment whose dimension scales as $\left(\varphi^{\frac{1}{6}}\right)^{L^2}$ seems more subtle since it contains neither inert spins nor unshaded plaquettes (that harbor athermal next-nearest neighbor spin correlations) in its bulk. However, initial states that arise from a wire pattern composed of $(2/3)L^2$ of the system in a checkerboard drum pattern, with a local density $n = 1/4$, and the rest of the system being fully inert with a local $n = 1/2$ also yields the same leading scaling of its fragment size as $\left(\varphi^{\frac{1}{6}}\right)^{L^2}$. Thus, it is clear that there is no single dominant Krylov subspace even at $n = 1/3$. We leave the issue of thermalization, or lack of it, or of an even more exotic feature like a behavior intermediate to both weak and strong Hilbert space fragmentation for the particular density of up-spins (bosons), $n = 1/3$, as an interesting open problem. For completeness, we note that ED results on small systems points towards a strong Hilbert space fragmentation scenario even at $n = 1/3$ (Fig. 11, right panel).

# 4 Analytical study of small quantum drums

In this section, we shall study the spectrum of some of the simplest quantum drums of the model analytically. We first discuss how the connection diagrams between different Fock states of a drum, where the connections are generated by $H$, can be represented by unidirectional trees in Sec. 4.1. Such tree structures turn out to be particularly useful in finding the non-zero matrix elements of $H$ for large drums (e.g., see Sec. 5). We will subsequently study the case of a wire with $N_p$ plaquettes for small $N_p$ (Sec. 4.2), and then consider some other examples of small quantum drums that can be viewed as building blocks of the different kinds of wire junctions shown in Fig. 5 (Sec. 4.3).

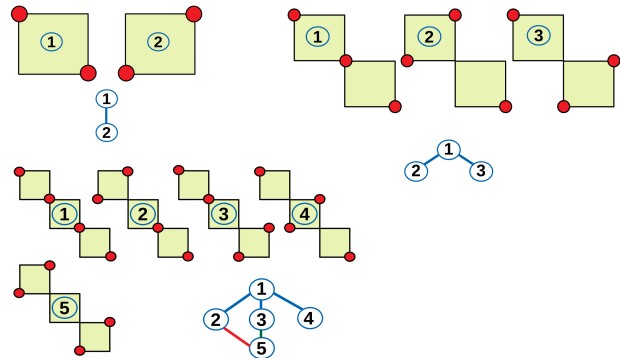

Figure 16: Top Left Panel:(a) Schematic representation of the basis states of the simplest fragment with $N_p = 1$ and (b) the corresponding tree between states in the Hilbert space. Top Right Panel: Same as the top left panel but corresponding to $N_p = 2$. Bottom Left Panel: Same as the top left panel but corresponding to $N_p = 3$. In all plots, the red filled dots indicates sites with up-spins (bosons). The green plaquettes in the drums follow the same convention as used in Fig. 1.

## 4.1 Tree structure

It is convenient to represent the connection diagram between different Fock states in the Hilbert space of a drum as nodes of a tree with the non-zero matrix elements of $H$ (which equals 1 due to the form of $H$ in Eq. 1) between two such states denoted as a link between the corresponding nodes. Such a tree can be build in the forward direction with the different levels being denoted by integers starting from level-0 for the top level and being incremented by 1 for each of the levels below. The top level consists of a single node that can be represented by any "reference state". It is optimal to choose a reference state such that the Fock state maximizes the number of flippable plaquettes for the corresponding drum, but this is not a necessary condition. A single application of $H$ on this reference state at level-0 generates all the nodes at level-1, where the corresponding Fock states have exactly 1 flipped plaquette with respect to the reference state with the location of the flipped plaquette uniquely identifying the corresponding Fock state. Links are then formed between nodes at level-0 and level-1. Applying $H$ on each of the level-1 nodes generates level-2 nodes where the corresponding Fock state has another flipped plaquette with respect to the level-1 state with the locations of the two flipped plaquettes characterizing the generated Fock state uniquely. The possibility that different level-$i$ nodes may generate the same level-$(i + 1)$ node first arises at $i = 1$. New links are then drawn between the appropriate nodes at level-1 and level-2. This process is continued recursively at each subsequent level-$i$ to go forward to level-$(i + 1)$ during which the links between appropriate level-$i$ and level-$(i + 1)$ nodes are also generated. Carrying out this forward construction of the tree, one also encounters "dead nodes" which are Fock states at level-$i$ from which no other Fock states with an extra flipped plaquette can be generated to go to the next level $(i + 1)$. The forward construction of the tree terminates when the last level is reached which is characterized by all its nodes being dead nodes. A plaquette, once flipped, cannot be unflipped in the tree construction which makes the construction unidirectional.

## 4.2 Wire

The simplest quantum drum of the Hamiltonian given by Eq. 1 constitutes a single plaquette ($N_p = 1$) with two up-spins (bosons) as shown in the top left panel of Fig. 16. The Hilbert

space of this drum constitutes two states; the Hamiltonian in the space of these two states, $|\psi_a\rangle \equiv |a\rangle$ for $a = 1, 2$, can be written as (Fig. 16)

$$H_{1\ell} = \tau^x, \qquad H_{1\ell}|\psi_{1(2)}\rangle = |\psi_{2(1)}\rangle, \tag{21}$$

where $\tau^x$ denotes Pauli matrix in the space of the states in the Hilbert space. This yields integer eigenvalues $E = \pm 1$.

The next set of quantum drums that we discuss constitutes two elementary square plaquettes ($N_p = 2$) with three up-spins (bosons) in total as shown in the top right panel of Fig. 16. The Hilbert space consists of three states, $|\phi_a\rangle \equiv |a\rangle$ for $a = 1, 2, 3$, as shown in the panel. The action of the Hamiltonian is summarized by the tree given in the top right panel of Fig. 16. In the space of these states, the Hamiltonian can be represented as

$$H_{2\ell} = \begin{pmatrix} 0 & 1 & 1 \\ 1 & 0 & 0 \\ 1 & 0 & 0 \end{pmatrix}. \tag{22}$$

This yields eigenvalues $E = 0, \pm\sqrt{2}$. Thus these fragments leads to eigenenergies which can be represented by simple irrational numbers as well as a zero mode.

Finally, we consider a wire with $N_p = 3$ where the states have 4 up-spins (bosons). The basis states spanning the 5-dimensional Hilbert space of such a fragment is charted in the bottom left panel of Fig. 16 and the corresponding tree is shown in the bottom panel of the figure. As can be read off from the tree, in the space of these states, the $5 \times 5$ Hamiltonian matrix can be written as

$$H_{3\ell} = \begin{pmatrix} 0 & 1 & 1 & 1 & 0 \\ 1 & 0 & 0 & 0 & 1 \\ 1 & 0 & 0 & 0 & 1 \\ 1 & 0 & 0 & 0 & 0 \\ 0 & 1 & 1 & 0 & 0 \end{pmatrix}. \tag{23}$$

The corresponding eigenvalues are given by $E = 0, \pm\sqrt{5 \pm \sqrt{17}}$ leading to eigenvalues represented by non-trivial irrational numbers and a zero mode. The spectrum of these wires for larger $N_p$ gets complicated and these shall be studied in details numerically in Sec. 5.

## 4.3 Junction units

In this section, we shall study small quantum drums corresponding to the simplest junction units that are building blocks of the different junctions of wires shown in Fig. 5 (A, B, C, D) and calculate their spectra analytically. Larger quantum drums that resemble a junction of two wires as shown in Fig. 5 (A) shall be studied numerically in Sec. 5.

We begin with the quantum drum corresponding to a junction of two wires as shown in Fig. 5 (A) with $N_p = 3$ elementary plaquettes. The basis states corresponding to such a junction is shown in the left panel of Fig. 17. The Hilbert space, as can be seen from this figure, is four dimensional. The tree for the states in the Hilbert space is shown in the bottom of the left panel Fig. 17. A straightforward analysis shows that $H$ admits a four-dimensional matrix representation in the space of these states which can be written in terms of outer product of two sets of Pauli and identity matrices ($\vec{\tau}_a$ and $I_a$ for $a = 1, 2$) as

$$H_{1j} = \tau_1^x \otimes (I_2 + \tau_2^z)/2 + I_1 \otimes \tau_1^x. \tag{24}$$

The corresponding eigenvalues satisfy the characteristic equation $E^4 - 3E^2 + 1 = 0$ and yields a solution $E = \pm(1 \pm \sqrt{5})/2$. These eigenvalues therefore yield the golden ratio for this particular drum.

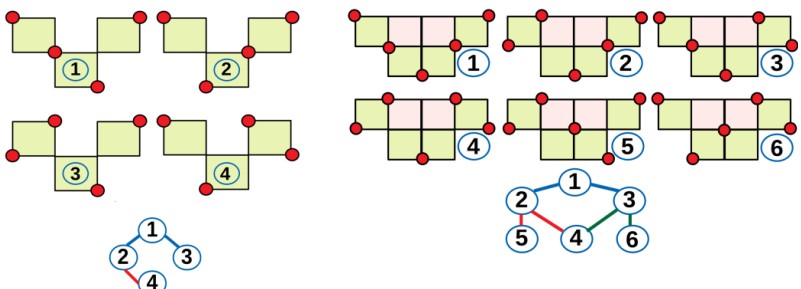

Figure 17: Left Panel:(a) Schematic representation of the basis states of the quantum drum corresponding to a junction of two wires as shown in Fig. 5 (A) with $N_p = 3$ and (b) the corresponding tree between the four states in the Hilbert space. Right Panel: Same as the left panel but corresponding to the simplest quantum drum (with $N_p = 4$) that can be treated as the junction unit to generate another junction of two wires as shown in Fig. 5 (B). In all plots, the red filled dots indicates sites with up-spins (bosons). The green (pink) plaquettes in the drums follow the same convention as used in Fig. 1.

Next, we consider the simplest quantum drum that can be treated as the junction unit to generate another junction of two wires as shown in Fig. 5 (B). This unit corresponds to a drum with $N_p = 4$ as shown in the right panel of Fig. 17. The basis states spanning the six-dimensional Hilbert space is shown in the top of the right panel of Fig. 17 while the tree for these states is shown in the bottom of this figure. We find that the Hamiltonian has a $6 \times 6$ matrix representation given by

$$
H_{3\ell} = \begin{pmatrix}
0 & 1 & 1 & 0 & 0 & 0 \\
1 & 0 & 0 & 1 & 1 & 0 \\
1 & 0 & 0 & 1 & 0 & 1 \\
0 & 1 & 1 & 0 & 0 & 0 \\
0 & 1 & 0 & 0 & 0 & 0 \\
0 & 0 & 1 & 0 & 0 & 0
\end{pmatrix}.
\tag{25}
$$

The characteristics equation for the eigenvalues simplifies to $E^2(E^2-1)(E^2-5) = 0$ and yields eigenvalues $E = 0$ (doubly degenerate) and $E = \pm 1, \pm\sqrt{5}$.

Next, we consider the simplest quantum drum that can be viewed as the junction unit of the junction of three wires shown in Fig. 5 (C). This drum has $N_p = 10$ plaquettes in it. The basis states spanning the Hilbert space is shown in the left panel of Fig. 18. The Hilbert space is 13 dimensional; the tree for these states is shown in the right panel of Fig. 18. This allows a $13 \times 13$ dimensional matrix representation of $H$. We do not write this matrix explicitly here since it can be easily constructed from the tree shown in Fig. 18. This matrix needs to be numerically diagonalized and yields eigenvalues $E = \pm 3.18259 \cdots, \pm 1.91182 \cdots, \pm(1 \pm \sqrt{5})/2, \pm 1, 0$, and $\pm 0.464856 \cdots$.

Finally, we study a quantum drum that is the junction unit of a junction of four wires as shown in Fig. 5 (D). This junction unit is shown in Fig. 19 and corresponds to $N_p = 16$ with nine up-spins (bosons) and 24 basis states. A few representative such states are shown in the left panel of Fig. 19. Each of these states belong to a different level in the tree starting from the state $|1\rangle$ as shown in the right panel of Fig. 19; they can be obtained from the state in the preceding level shown by application of $H$. The other states can be analogously obtained following the tree; we do not show them explicitly to avoid clutter. The tree shows that $H$ admits a $24 \times 24$ matrix representation. Remarkably, this matrix can be analytically diagonalized; its

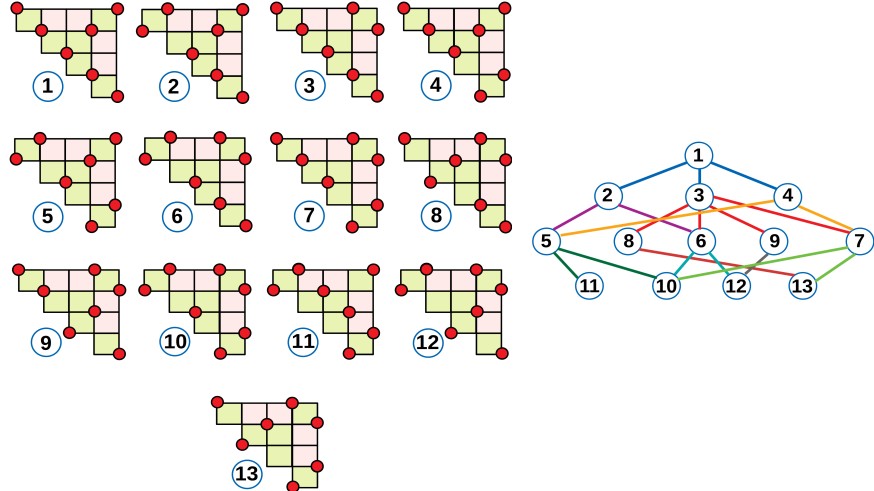

Figure 18: Left Panel: Schematic representation of the basis states of the simplest quantum drum that can be viewed as the junction unit of the junction of three wires shown in Fig. 5 (C) with $N_p = 10$ plaquettes. The red filled circles indicates sites with up-spins (bosons). Right Panel: The corresponding tree between the 13 states in the Hilbert space. The green (pink) plaquettes in the drum follows the same convention as used in Fig. 1.

eigenvalues satisfies the characteristics equation

$$E^6(E^2 - 2E - 2)(E^2 - 2)(E^4 - 22E^2 + 80)(E^4 - 6E^2 + 6)^2(E^2 + 2E - 2) = 0. \qquad (26)$$

These leads to the 24 eigenvalues given by 0 (six fold degenerate), $\pm\sqrt{3 \pm \sqrt{3}}$ (each two fold degenerate), $\pm\sqrt{11 \pm \sqrt{41}}$, $\pm\sqrt{2}$, and $\pm 1 \pm \sqrt{3}$.

## 5 Numerical study of two quasi-1D quantum drums

In this section, we will numerically calculate the spectrum of large quantum drums with $N_p$ elementary plaquettes using the examples of a wire (Fig. 20, left and middle panels have $N_p = 4$ and 5, respectively) and a particular junction of two equal length wires (Fig. 20, right panel with $N_p = 7$). We refer to this latter case as "junction of two wires" henceforth. Using exact diagonalization (ED), we could calculate the spectrum up to $N_p = 22$ for the wire and $N_p = 23$ for the junction of two wires. We will show that the spectrum of a wire with $N_p$ plaquettes is identical to the paradigmatic 1D PXP chain [12, 13] with $N_p$ sites on a chain with OBC. This equivalence allows us to extract several features of the high-energy spectrum of the wire from known results in the literature [12, 13]. However, our numerical studies also reveal enhanced fidelity revivals from a period-3 initial state for $N_p = 3n + 1$, where $n$ is an integer, without the need of adding any optimal perturbations to the Hamiltonian which was not pointed out earlier in the literature. While the junction of two wires differs from the wire by only a "surface term" when $N_p$ is large, the structure of the Hilbert space is completely different and gives a different constrained model compared to the 1D PXP chain. Thus, the presence or absence of a single junction leads to interesting differences in the high-energy spectrum that persist for large drums.

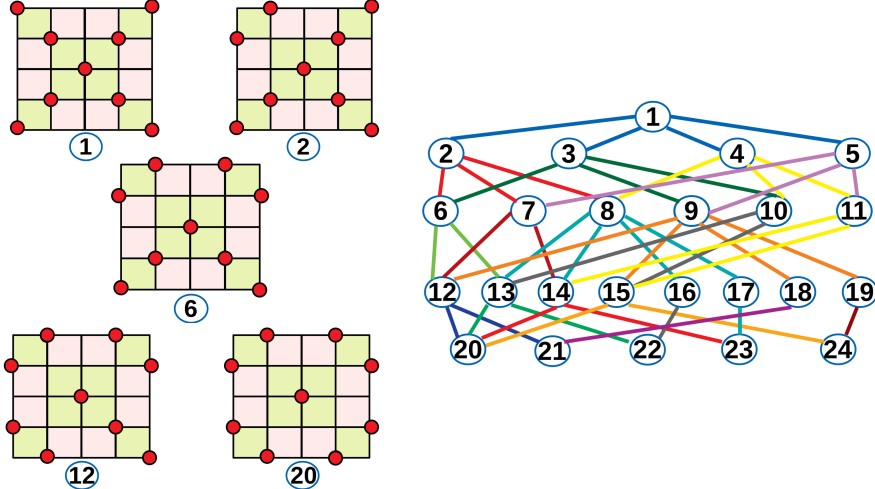

Figure 19: Left Panel: Schematic representation of a few basis states of the drum that can be viewed as the junction unit of a junction of four wires as shown in Fig. 5 (D). The drum contains $N_p = 16$ plaquettes. The red filled circles indicates sites with up-spins (bosons). Right Panel: The corresponding tree between the 24 states in the Hilbert space. Each of the states can be obtained from a state connected to it by application of $H$ on it. The green (pink) plaquettes in the drum follows the same convention as used in Fig. 1. See text for details.

## 5.1 Tree generating algorithm and equivalence of wire to 1D PXP chain

The concept of a unidirectional tree starting from a reference state has already been introduced in Sec. 4.1. The reference state for a wire with $N_p$ plaquettes can be taken to be the Fock state with all $N_p + 1$ up-spins to be along the wire length as previously done in Sec. 2.4. For the junction of two wires with $N_p = 2x + 1$ plaquettes, we take the reference state to be the one where $x + 2\,[x]$ up-spins (bosons) are arranged along the length of the wire of length $x + 1\,[x]$ to the right [left] of the central junction plaquette, including [excluding] the junction plaquette (e.g., see an example of the reference state marked as ① in the left panel of Fig. 22.). For both the wire and the junction of two wires, computationally it is convenient to adopt a one-to-one map from a spin configuration on the wire or a junction of two wires to another defined on an open chain with $N_p$ sites in 1D where each site of the chain can have a pseudospin variable $\tau_i^z = \pm 1$ or $0$, where $i = 1$ to $N_p$. The pseudospins on the chain represent the plaquettes of the drum sequentially from left to right in both the cases. For the wire, these variables take the value $+1$ $(-1)$ for elementary plaquettes that have two up-spins along (perpendicular to) the wire direction and $0$ otherwise (Fig. 21, top panel). For the junction of two wires, we follow the same convention and remove the ambiguity at the central plaquette by associating it to the wire to the right of the junction plaquette (Fig. 22, left panel). While a pseudospin with $0$ has multiple possibilities associated with an elementary plaquette involving states with zero or one up-spin, specifying the locations of the $\pm 1$ pseudospins also fixes the spin state of the other plaquettes on the drum.

The tree generating algorithm then proceeds as follows. One starts with the reference state which has $\tau_i^z$ equal to $111\cdots 1$ for the wire and $11\cdots 1011\cdots 1$ for the junction of two wires where the $0$ in the latter case represents the plaquette to the immediate left of the 1-junction plaquette. For the wire, the states at subsequent levels are generated by flipping a $1$ to $-1$ and replacing the pseudospins at neighboring site(s) of the flipped pseudospin by $0$ (Fig. 21). For the junction of two wires, the rules are practically the same except at the 1-junction plaquette

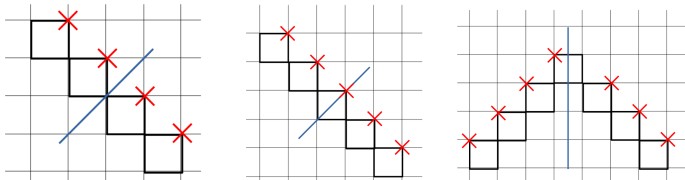

Figure 20: Two wires with $N_p = 4$ (left panel) and $N_p = 5$ (middle panel) respectively and a junction of two wires with $N_p = 7$ (right panel) are illustrated here. These quantum drums have a discrete reflection symmetry with the thick blue line in all the three panels indicating the corresponding axis of reflection. A chiral operator can be constructed from the product of $\sigma^z$ on all sites indicated by $\times$ (in red) for these drums.

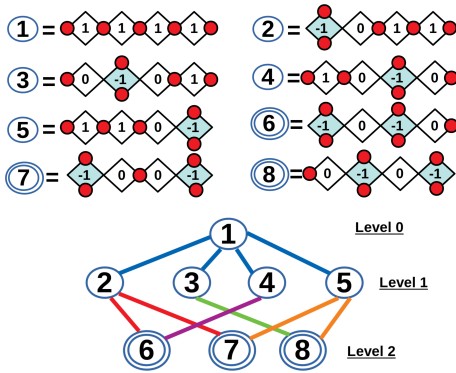

Figure 21: Tree structure (bottom panel) for a wire with $N_p = 4$ shown here. The $i$th level contains Fock states with $i$ flipped plaquettes (indicated by shaded plaquettes in the top panel) with respect to the reference state defined in level-0. The nodes between two Fock states imply that these are connected with a single application of $H$. The Fock states enclosed by double circles represent dead ends of the tree. The corresponding value of the pseudospin variable $(\pm 1, 0)$ is also shown at the center of each plaquette for each of the Fock states in the top panel.

denoted by the site $i_0$ on the open chain. When 1 is flipped to $-1$ at $i_0$, while the pseudospin at $i_0 + 1$ is replaced by 0 as usual, the pseudospin at $i_0 - 1$ is replaced by 0 if the pseudospin at $i_0 - 2$ equals $-1$, else it is replaced by $+1$ (Fig. 22). Following this algorithm, we generate the tree and the corresponding $H$ matrix for both the quantum drums being discussed here.

We now show that the wire with $N_p$ plaquettes has the same spectrum as that of the 1D PXP chain with $N_p$ sites and OBC, whose Hamiltonian is defined as follows:

$$H_{\text{PXP}} = \sum_{i=2}^{N_p-1} P_{i-1} \mu_i^x P_{i+1} + \mu_1^x P_2 + P_{N_p-1} \mu_{N_p}^x, \tag{27}$$

where $\mu_i^\alpha$ for $\alpha = x, y, z$ represents spin-1/2 Pauli matrices at site $i$ of the open chain with $N_p$ sites, and $P_i = (1 - \mu_i^z)/2$ is a local projection operator. The constrained Hilbert space of the PXP chain is defined by the condition that no two nearest neighbor sites $i, i + 1$ can have $\mu_i^z = +1$ and $\mu_{i+1}^z = +1$ together. We now make the following correspondence between the

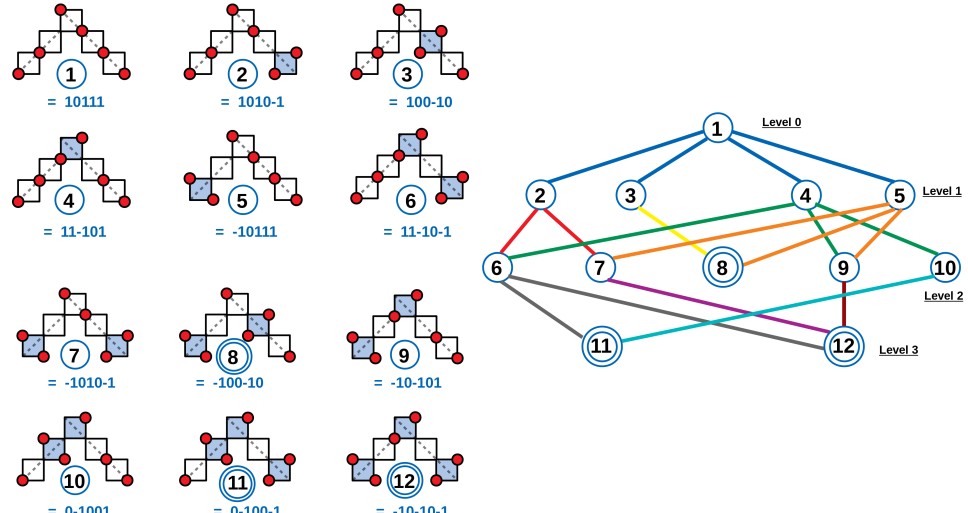

Figure 22: Tree structure for a junction of two wires with $N_p = 5$ shown here. The $i$th level contains Fock states with $i$ flipped plaquettes (indicated by shaded plaquettes) with respect to the reference state defined in level-0. The nodes between two Fock states (where the links imply the forward construction of the tree) imply that these are connected with a single application of $H$. The Fock states enclosed by double blue circles represent dead ends of the tree. The pseudospins on an open chain with $N_p = 5$ sites is also shown for each of the Fock states in the left panel where red filled circles denote up-spins (bosons).

pseudospins $\tau_i^z$ for the wire and the spins $\mu_i^z$ for the PXP chain:

$$\begin{aligned}
\tau_i^z = -1 &\Rightarrow \mu_i^z = +1\,, \\
\tau_i^z = +1 &\Rightarrow \mu_i^z = -1_{\mathrm{f}}\,, \\
\tau_i^z = 0 &\Rightarrow \mu_i^z = -1_{\mathrm{uf}}\,,
\end{aligned} \tag{28}$$

where $\mu_i^z = -1_{\mathrm{f}}$ ($-1_{\mathrm{uf}}$) implies that flipping $\mu_i^z$ from $-1$ to $+1$ is allowed (disallowed) due to the hard-core constraints of the 1D PXP chain. In this language, the reference state of the wire with $\tau_i^z = +1$ for all $i$ corresponds to the "Rydberg vacuum" state of the PXP chain with no Rydberg excitations, i.e., $\mu_i^z = -1_{\mathrm{f}}$ for all $i$. The tree generating algorithm then constructs a unidirectional tree starting from the reference state at level-0 by flipping a $\tau_i^z = 1$ to $\tau_i^z = -1$ and replacing the pseudospins at neighboring site(s) of the flipped pseudospin by $\tau_{i+1}^z = \tau_{i-1}^z = 0$ for $i \neq 1, N_p$ and $\tau_{i+1}^z = 0$ ($\tau_{i-1}^z = 0$) for $i = 1$ ($i = N_p$) at each subsequent level of the tree. The action of $H_{\mathrm{PXP}}$ in Fock space can also be represented by the same tree structure as the wire using Eq. 28 since flipping any $\mu_i$ from $-1$ to $+1$ starting from the Rydberg vacuum state automatically makes the previously flippable nearest neighbor site(s) with $\mu = -1$ unflippable due to the hard-core constraints of the PXP chain.

We note that this equivalence immediately breaks down for the junction of two wires since flipping a pseudospin $\tau_{i_0}^z = +1$ to $\tau_{i_0}^z = -1$ on the central junction plaquette, denoted by $i_0$, starting from the reference state produces a flippable $\tau_{i_0-1}^z = +1$ to its immediate left (see Fock states marked by 1 and 4 in the left panel of Fig. 22 for an example) which implies that the junction of two wires cannot be represented by the same constrained Hilbert space as the PXP chain by this mapping.

## 5.2 Hilbert space dimension and level statistics

Let us calculate the Hilbert space dimension for both these drums for an arbitrary $N_p$ which will justify their interpretation as effective quasi-1D models since the dimensionality scales exponentially with $N_p$ as $N_p \gg 1$ in both cases. Let us denote the number of possible Fock states for a wire with $N_p$ plaquettes to be $\mathcal{N}_w(N_p)$. All the Fock states for such a wire can be built in either one of the following two ways. Consider starting from the reference state (Fig. 21, level-0 state) and building all possible Fock states using the first $N_p - 1$ plaquettes starting from the top. The number of generated states then equals $\mathcal{N}_w(N_p - 1)$ and it is easy to see that the last plaquette will then either have the pseudospin to be $+1$ or $0$. The remaining states of the wire with $N_p$ plaquettes can be generated by starting from the reference state and fixing the pseudospin of the last plaquette to be $-1$ (i.e., flipping this last plaquette). The first $N_p - 2$ plaquettes from the top can then be used to generate the missing Fock states whose number equals $\mathcal{N}_w(N_p - 2)$. Thus, we get that

$$\mathcal{N}_w(N_p) = \mathcal{N}_w(N_p - 1) + \mathcal{N}_w(N_p - 2) = F_{N_p + 2}. \tag{29}$$

By construction, $\mathcal{N}_w(1) = 2$ and $\mathcal{N}_w(2) = 3$ which implies that $\mathcal{N}_w(N_p) = F_{N_p + 2}$ as written above, where $F_n$ are the Fibonacci numbers defined by the recurrence relation $F_0 = 0$, $F_1 = 1$ and $F_n = F_{n-1} + F_{n-2}$ for $n > 1$.

Similarly, for the junction of two wires with $N_p = 2x + 1$ plaquettes, all the Fock states can again be built in one of the following two ways. Consider starting from the reference state (Fig. 22, level-0 state) and building all possible Fock states of the left wire with $x - 1$ plaquettes starting from the left-bottom plaquette and the right wire with $x + 1$ plaquettes starting from the right-bottom plaquette. The number of such states equal $\mathcal{N}_w(x-1)\mathcal{N}_w(x+1)$ and the plaquette to the immediate left of the central junction plaquette can have a pseudospin of either be $+1$ or $0$. To generate the remaining configurations, we start from the reference state again and make the pseudospin of this particular plaquette to be $-1$ by first flipping the central junction plaquette and then flipping the plaquette to the immediate left of the junction. The number of Fock states generated from the rest of the plaquettes then equals $\mathcal{N}_w(x-1)\mathcal{N}_w(x-2)$, thus giving the relation

$$\mathcal{N}_j(N_p = 2x + 1) = \mathcal{N}_w(x-1)[\mathcal{N}_w(x-2) + \mathcal{N}_w(x+1)] = F_{x+1}(F_x + F_{x+3}), \tag{30}$$

where $\mathcal{N}_j(N_p = 2x + 1)$ refers to the number of Fock states in a junction of two wires composed of $N_p = 2x + 1$ elementary plaquettes. Eq. 29 and Eq. 30 show that the number of allowed Fock states scale exponentially for large $N_p$ for both the drums. Note that while Eq. 29 is identical to the Hilbert space dimension of a 1D PXP chain with $N_p$ sites and OBC, as should be the case from the equivalence of both models shown in Sec. 5.1, the Hilbert space dimension of the junction of two wires (Eq. 30) cannot be expressed as $F_m$ with an integer $m$ in general showing that the structure of the constrained Hilbert space of this drum is different from that of the 1D PXP chain.

We can then ask whether these large quantum drums satisfy a Krylov-restricted version of the ETH, i.e., whether these quasi-1D models are non-integrable. We check this using the method of level statistics that can be obtained directly using the eigenspectrum from ED (e.g., see Ref. [79]). To calculate the level statistics for large quantum drums, it is important to first project to a sector where all the commuting global symmetries have been resolved. Since both the wire and the junction of two wires (Fig. 20) are quasi-1D structures with open boundaries, momentum is not a good quantum number. The total magnetization in the computational basis, $S_{\text{tot}}^z$, represents a conserved quantity for these drums. However, all nodes of a tree (Fig. 21 and Fig. 22) already have the same $S_{\text{tot}}^z$ by construction. The only remaining non-trivial global symmetry turns out to be a reflection symmetry, denoted by $\mathcal{R}_w$ ($\mathcal{R}_j$) for the

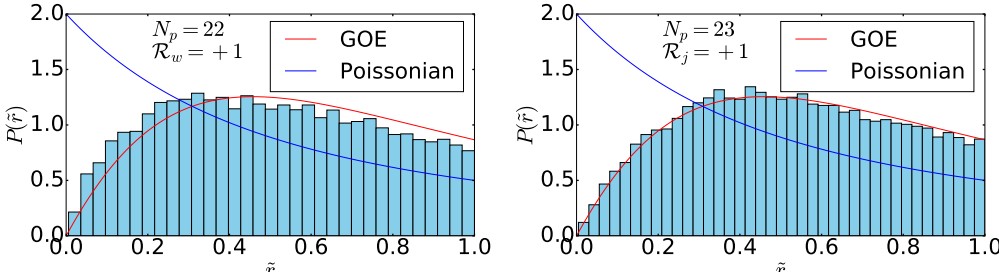

Figure 23: Level spacing ratio distribution $P(\tilde{r})$ versus $\tilde{r}$ for a wire with $N_p = 22$ (left panel) and a junction of two wires with $N_p = 23$ (right panel), with both the data taken in the symmetry sector with $\mathcal{R}_{w/j} = +1$. The histograms indicate the non-integrability of both the quasi-1D models.

wire (junction of two wires), which takes a Fock state $|\alpha\rangle$ to another Fock state $|\beta\rangle = \mathcal{R}_{w/j}|\alpha\rangle$ with the axis of reflection shown in Fig. 20 for both the drums. For a wire with even (odd) number of plaquettes, the axis passes through a site (the diagonal of a square) (Fig. 20, left and middle panels) whereas for a junction of two wires, it passes through the central junction plaquette as shown in Fig. 20, right panel. Since $\mathcal{R}_{w/j}^2|\alpha\rangle = |\alpha\rangle$ for any Fock state, the basis states $(|\alpha\rangle \pm \mathcal{R}_{w/j}|\alpha\rangle)/\sqrt{2}$ define states with $\mathcal{R}_{w/j} = \pm 1$ respectively. If $\mathcal{R}_{w/j}|\alpha\rangle = |\alpha\rangle$ for some Fock state(s), then such Fock state(s) only contribute to the $\mathcal{R}_{w/j} = +1$ sector. This happens in the case of the wire, where, the reference state provides one example of such a Fock state. Thus, the number of basis states in $\mathcal{R}_w = +1$ always exceeds the corresponding number for $\mathcal{R}_w = -1$ for a wire whereas these two numbers are equal to each other for a junction of two wires.

Restricting to the larger sector with $\mathcal{R}_{w/j} = +1$, we construct the distribution of consecutive level spacing ratios $\tilde{r}$ (with support in $[0,1]$) where $\tilde{r}$ is defined as follows:

$$\tilde{r} = \min\left\{r_n, \frac{1}{r_n}\right\} \leq 1, \quad r_n = \frac{s_n}{s_{n-1}}, \qquad s_n = E_{n+1} - E_n, \tag{31}$$

where $E_n$ represent the energies of the eigenvectors obtained from ED. For a non-integrable model, one expects a Gaussian orthogonal ensemble (GOE) distribution, while an integrable system leads to a Poisson distribution for $P(\tilde{r})$ [80], where the two distributions have the following forms:

$$P_{\text{GOE}}(\tilde{r}) = \frac{27}{4}\frac{\tilde{r} + \tilde{r}^2}{(1 + \tilde{r} + \tilde{r}^2)^{5/2}}, \quad P_{\text{P}}(\tilde{r}) = \frac{2}{(1 + \tilde{r})^2}. \tag{32}$$

The numerically generated data for $P(\tilde{r})$ versus $\tilde{r}$ is shown for a wire with $N_p = 22$ plaquettes and a junction of two wires with $N_p = 23$ plaquettes in Fig. 23. The data clearly indicates that $P(\tilde{r})$ follows $P_{\text{GOE}}(\tilde{r})$ much more closely than $P_{\text{P}}(\tilde{r})$ for these system sizes giving strong evidence for the non-integrable nature of both these quasi-1D models.

## 5.3 Zero modes and index theorem

While both the wire and the junction of two wires have a symmetric eigenspectrum of $H$ around $E = 0$ as is expected for any quantum drum, the ED data further reveals the presence of an ever-increasing number of exact zero modes (up to machine precision) with increasing $N_p$ for the former case and the absence of any zero mode for the latter case.

This striking difference between the two drums can be understood in terms of the index theorem of Ref. [75]. Firstly, a chiral operator $\mathcal{C}_{w/j} = \prod_{\square_j}\sigma_{j_x,j_y}^z$ (where the subscript $w(j)$

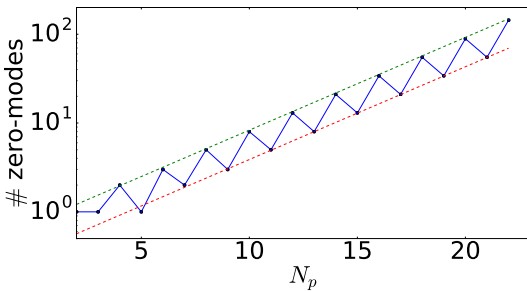

Figure 24: Scaling of the total number of zero modes versus $N_p$ for a wire. For even (odd) values of $N_p$, the number of zero modes grow as $\mu_e(\sqrt{\varphi})^{N_p}$ ($\mu_o(\sqrt{\varphi})^{N_p}$) where $\mu_e \approx 0.75$ ($\mu_o \approx 0.35$) [indicated by dotted lines].

refers to the wire (junction of two wires)) can be defined in both cases, which involves one site $(j_x, j_y)$ per elementary plaquette contained in the drum (these sites are indicated by crosses in red in all panels of Fig. 20). This operator satisfies $\{H, \mathcal{C}_{w/j}\} = 0$ for the Hilbert space fragments generated by these drums, thus ensuring the $E \rightarrow -E$ symmetry of the spectrum. Furthermore, as already discussed, these two drums have a global reflection symmetry, $\mathcal{R}_{w/j}$, that commutes with $H$ (Fig. 20). Importantly, while $[\mathcal{R}_w, \mathcal{C}_w] = 0$, it turns out that $[\mathcal{R}_j, \mathcal{C}_j] \neq 0$ which means that the index theorem of Ref. [75] applies to the wire but not to junction of two wires. This leads to a macroscopically large number of protected zero modes in the former case and also explains our numerical data (Fig. 24). The number of zero modes in the wire show an interesting even-odd effect as a function of $N_p$ (Fig. 24) with the even values of $N_p$ showing a higher number of zero modes. This even-odd effect stems from the fact that the axis that defines the reflection symmetry, $\mathcal{R}_w$, passes through a single site shared by two elementary plaquettes for even values of $N_p$; in contrast, it passes through two sites along a diagonal of an elementary plaquette for odd values of $N_p$ (Fig. 20, left and middle panels). The number of zero modes scale as $\mu_{e/o}(\sqrt{\varphi})^{N_p}$ (with $\varphi = (1 + \sqrt{5})/2$ being the golden ratio as defined before) where $\mu_e \approx 0.75$ ($\mu_o \approx 0.35$) for even (odd) values of $N_p$ (see Fig. 24). Identical scaling behavior was also observed for the number of zero modes in the 1D PXP model [12, 13, 81].

It is useful to point out here that a different type of junction of two equal-length wires (Fig. 5, panel B) instead of this junction being studied here will again have an exponentially large number of exact zero modes. Similarly, a junction of three equal-length wires (Fig. 5, panel C) as well as a junction of four equal-length wires (Fig. 5, panel D) will also have a macroscopic number of zero modes due to the index theorem of Ref. [75].

## 5.4 QMBS and related diagonastics

While the level statistics distribution of both the wire and the junction of two wires (Fig. 23) is consistent with these quasi-1D models being non-integrable for large $N_p$ and thus satisfying Krylov-restricted ETH, both quantum drums also harbor QMBS that give rise to observable dynamical signatures like periodic revivals from certain simple initial states.

Let us consider three such Fock states for the wire as shown in Fig. 25. Fig. 25 (left panel) shows the reference state which we denote as $|r\rangle_w$, Fig. 25 (middle panel) shows a Fock state obtained by flipping every alternate elementary plaquette in the reference state which we denote as $|fu\rangle_w$, and Fig. 25 (right panel) shows a Fock state obtained by flipping every third elementary plaquette in the reference state which we denote as $|fuu\rangle_w$. Similarly, two representative Fock states are shown in Fig. 26 for the junction of two wires. Fig. 26 (left panel)

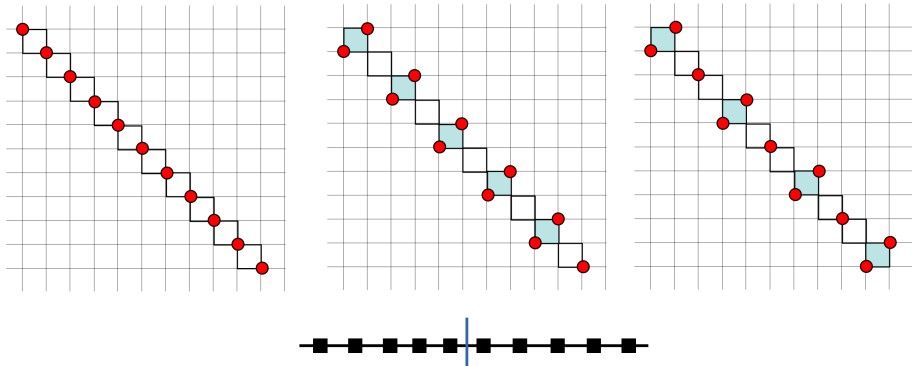

Figure 25: Three Fock states shown for the wire with $N_p = 10$. (Left panel) Reference state denoted by $|r\rangle_w$ (Middle panel) A $|fu\rangle_w$ state created by flipping every alternate elementary plaquette in the reference state (Right panel) A $|fuu\rangle_w$ state created by flipping every third elementary plaquette of the reference state. Flipped plaquettes with respect to the reference state are shown shaded in blue. The entanglement cut used to calculate the bipartite entanglement entropy of the system after mapping it to an open chain of pseudospins is shown below; such a cut divides the system into two equal halves with $N_p/2$ pseudospins each.

shows the reference state which we denote as $|r\rangle_j$ (note that there are two such reference states possible for the junction of two wires) and Fig. 26 (right panel) shows a Fock state obtained by flipping every alternate elementary plaquette in the reference state which we denote as $|fu\rangle_j$. From the equivalence of the wire to the 1D PXP chain shown in Sec. 5.1, it is clear that while local operators starting from the state $|r\rangle_w$ will thermalize quickly, since the initial state maps to the Rydberg vacuum state of the PXP chain, this will not be the case from the initial states $|fu\rangle_w$ and $|fuu\rangle_w$ which map to the period-2 $|\mathbb{Z}_2\rangle$ and the period-3 $|\mathbb{Z}_3\rangle$ Fock states of the PXP chain, respectively.

This can indeed be checked by monitoring the fidelity $F(t) = |\langle s| \exp(-iHt)|s\rangle|^2$ using ED for these representative Fock states (denoted by $|s\rangle$) in Fig. 27. Most initial states show a rapid drop in $F(t)$ within $t \sim O(1)$ which is expected for a high-energy initial state in an interacting system. However, for the wire, the behaviour of $F(t)$ for $|fu\rangle_w$ and $|fuu\rangle_w$ are markedly different, with both showing periodic revivals with an emergent time-scale $T^* \sim 5$ for $|fu\rangle_w$ (Fig. 27, top-left panel) and $T^* \sim 4$ for $|fuu\rangle_w$ (Fig. 27, top-right panel). The periodic revivals of $F(t)$ starting from $|fu\rangle_w$ show a decaying envelope in time that can be reasonably described by the envelope function $\exp(-t/\tau_w)$ with $\tau_w \approx 10$ (Fig. 27, top-left panel). This decaying envelope to the periodic revivals distinguish this phenomenon from the persistent oscillations starting from initial Fock states discussed in Sec. 2.3. In fact, the fidelity revivals for the $|fuu\rangle_w$ state shows a very interesting finite-size effect with such revivals being strongest when $N_p = 3n + 1$ where $n$ is an integer. For example, the peak value of the first fidelity revival in time equals 0.52 for $N_p = 19$ while it is much smaller for $N_p = 18$ and $N_p = 20$ (0.22 and 0.18 respectively). Furthermore, the $N_p = 19$ data for $F(t)$ starting from the initial state $|fuu\rangle_w$ shows no sign of a decaying exponential envelope till $t = 50$ (see Fig. 27, top-right panel). For the junction of two wires with $N_p = 23$, we again see rapid decay of $F(t)$ starting from $|r\rangle_j$ (Fig. 27, bottom-left panel) while $F(t)$ shows non-trivial periodic revivals from $|fu\rangle_j$ with the same $T^* \sim 5$ as in the wire case. The periodic revivals are weaker for the junction of two wires compared to the single wire and again have a decaying exponential envelope described by $\exp(-t/\tau_j)$ with a smaller $\tau_j \approx 7$, but these fidelity revivals are nonetheless clearly visible up to $t \sim 20$.

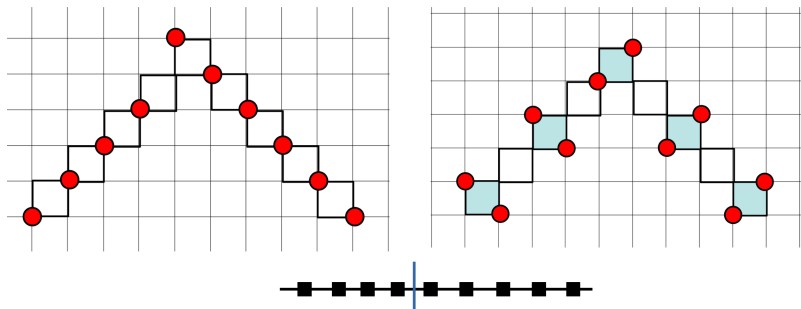

Figure 26: Two Fock states shown for the junction of two wires with $N_p = 9$. (Left panel) Reference state denoted by $|r\rangle_j$ (Right panel) A $|fu\rangle_j$ state created by flipping every alternate elementary plaquette in the reference state. Flipped plaquettes with respect to the reference state are shown shaded in blue. The entanglement cut used to calculate the bipartite entanglement entropy of the system after mapping it to an open chain of pseudospins divides the system into two halves with $(N_p/2) \pm 1$ pseudospins as shown below.

It is useful to point out that the enhanced revivals observed for $|fuu\rangle_w$ for $N_p = 3n + 1$ for the wire, which is equivalent to a $|\mathbb{Z}_3\rangle$ initial state in a 1D PXP chain with $N_p$ sites and OBC, was not pointed out in the literature previously and additional terms were added to the PXP Hamiltonian to cause enhancement of fidelity revivals from the $|\mathbb{Z}_3\rangle$ state [82]. As is well-known from the 1D PXP chain [12,13], these fidelity revivals from certain special initial states is due to a large overlap with *approximate* towers of QMBS that are equally spaced in energy. These towers are most clearly seen by plotting the overlaps of the initial Fock state $|fuu\rangle_w$ with the many-body eigenstates $|E\rangle$ as a function of energy (see Fig. 28). We see that at $N_p = 3n + 1$ (Fig. 28, middle panel), these towers are much more clearly formed compared to $N_p = 3n$ (Fig. 28, left panel) and to $N_p = 3n + 2$ (Fig. 28, right panel). We also note that at the system sizes, $N_p = 3n + 1$, the $|fuu\rangle_w$ Fock state becomes orthogonal to the zero mode subspace of the system (up to machine precision) even though the initial state has zero average energy. A deeper understanding of all these striking finite-size effects at $N_p = 3n + 1$ for the wire/open PXP chain would be highly desirable.

Even though the junction of two wires cannot be reduced to the 1D PXP chain, this model also admit *approximate* towers of QMBS that are equidistant in energy. In Fig. 29 (two panels), the overlap behavior of the $|fu\rangle_j$ Fock state (Fig. 26, right panel) with the eigenstates of the junction of two wires with $N_p = 23$ is shown for the $\mathcal{R}_j = +1$ and the $\mathcal{R}_j = -1$ sectors respectively. In this case, the towers of states with higher overlap to the Fock state are somewhat less clearly separated from the bulk of the spectrum as compared to the 1D PXP model, explaining the weaker fidelity revivals in the junction of two wires as compared to the single wire case (Fig. 27, top left and bottom right panels). Since $[\mathcal{R}_j, \mathcal{C}_j] \neq 0$, the overlaps are not symmetric with respect to zero energy; rather, the overlap behavior for $\mathcal{R}_j = +1$ sector is a *mirror image* (with the mirror axis being $E = 0$) of the $\mathcal{R}_j = -1$ sector.

Another tell-tale signature for the presence of QMBS is that such states have anomalously low bipartite entanglement entropy compared to neighboring eigenstates with similar energies. The bipartite entanglement entropy is given by

$$S(A) = -\mathrm{Tr}[\rho_A \ln \rho_A], \tag{33}$$

for each eigenstate $|\Psi\rangle$ where $\rho_A = \mathrm{Tr}_{\bar{A}}|\Psi\rangle\langle\Psi|$ where $\rho_A$ represents the reduced density matrix obtained by partitioning the system in to two spatial regions, $A$ and its complement $\bar{A}$. We find it convenient to compute the bipartite entanglement entropy by adopting the one-to-one

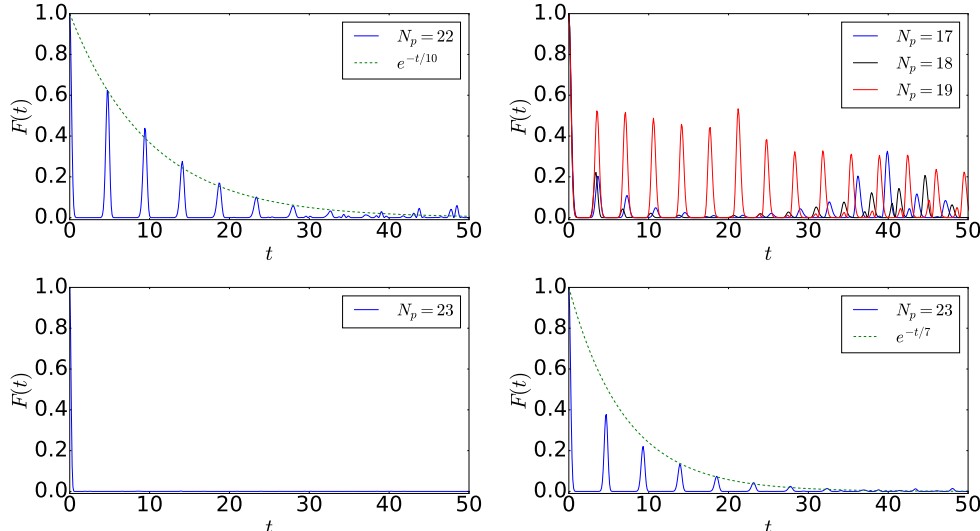

Figure 27: The behavior of fidelity $F(t)$ shown for the wire with two different initial Fock states with the top-left panel for $|fu\rangle_w$, and the the top-right panel for $|fuu\rangle_w$. The bottom-left (bottom-right) panel shows the fidelity as a function of time with the initial state being $|r\rangle_j$ ($|fu\rangle_j$) for a junction of two wires.

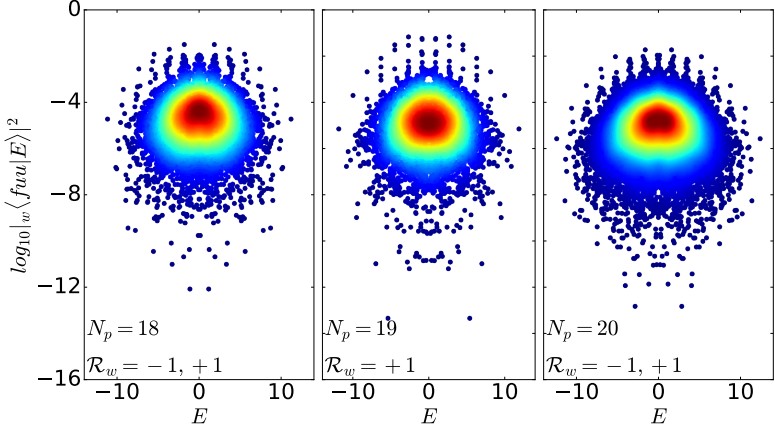

Figure 28: Density plots showing the overlap of the $|fuu\rangle_w$ state with energy eigenstates of the wire with $N_p = 18$ (left panel), $N_p = 19$ (middle panel) and $N_p = 20$ (right panel) respectively. In all the panels, the density of states is indicated by the same color map where warmer color corresponds to higher density of states.

mapping of Fock states in a wire or a junction of two wires to pseudospins with values $0, \pm 1$ in a 1D open chain with $N_p$ sites, with the mapping explained in Sec. 5.1. We then take $\bar{A}$ to be the first $N_p/2$ sites of the 1D chain for the wire (as shown in Fig. 25) and the first $(N_p/2) - 1$ sites of the 1D chain for the junction of two wires (as shown in Fig. 26). The results of such a computation from ED are shown in Fig. 30 for the wire (top-left panel) and the junction of two wires (top-right panel) respectively. While both the panels show a presence of several anomalous eigenstates with lower bipartite entanglement entropy than the bulk of the spectrum, the wire shows a broader distribution of values especially in the neighborhood of $E = 0$ compared to the junction of two wires.

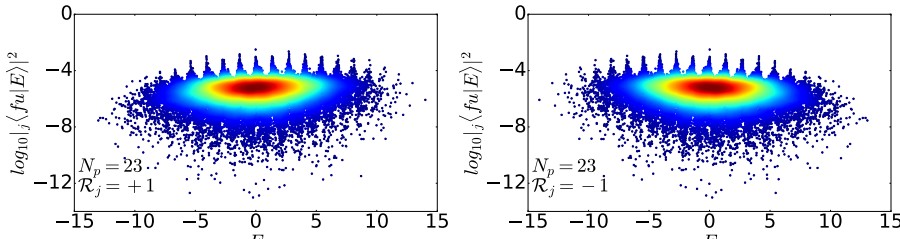

Figure 29: Density plots that show the overlap of the $|fu\rangle_j$ state for a junction of two wires with energy eigenstates of the junction with two wires with $N_p = 23$ for the $\mathcal{R}_j = +1$ (left panel) and the $\mathcal{R}_j = -1$ (right panel) sectors respectively. In both panels, the density of states is indicated by the same color map where warmer color corresponds to higher density of states.

The expectation value of any local operator in a high-energy eigenstate is supposed to approach the thermal result, with the inverse temperature being fixed by the energy density of the eigenstate, for a system that satisfies ETH. In Fig. 30 (bottom panels), we consider the expectation value $\langle\Psi|\mathcal{O}|\Psi\rangle$ as a function of energy, where $|\Psi\rangle$ denotes an eigenstate of the wire (bottom left panel) or the junction of two wires (bottom right panel) and $\mathcal{O} = \sigma_1^z\sigma_3^z + \sigma_2^z\sigma_4^z$ where sites $1, 2, 3, 4$ represent the four sites (in a clockwise manner) of the $N_p/2$-th elementary plaquette from the top-left for a wire and the central junction plaquette for the junction of two wires. Since this local operator is located away from the edges of the system, it represents a bulk operator in both the cases. The thermal result as a function of energy is represented by dotted curves on both the lower panels of Fig. 30. While the expectation value of the local operator for the bulk of the spectrum indeed approaches the thermal result, several eigenstates do show an expectation value that is quite far from the corresponding thermal result. The wire again shows a much larger variation in the range of expectation values compared to the junction of two wires, especially in the vicinity of $E = 0$. Interestingly, the latter case shows tower-like structures that are equidistant in energy (Fig. 30 (bottom right panel)) with a similar spacing between them as the tower of scar states visible in the overlap plots (Fig. 28, two panels).

## 6 Discussion

In conclusion, we have considered a spin-1/2 model on the two-dimensional square lattice in a constrained Hilbert space where no two nearest-neighbor sites can have up-spins simultaneously. The interaction Hamiltonian is composed of ring-exchange terms on elementary plaquettes that not only conserve the total magnetization but also the magnetization along each column and row of the square lattice. These additional subsystem symmetries imply conservation of a global dipole moment that leads to the phenomenon of Hilbert space fragmentation. While microscopic models of both weak and strong fragmentation are known in one dimension, we show that this particular interacting model with both hard-core constraints and subsystem symmetries presents a rich structure of emergent quantum drums as well as a rare example of strong Hilbert space fragmentation in two dimensions.

All the many-body eigenstates of this model can be expressed in terms of the tensor product of modes of appropriate quantum drums and any left-over inert spins. Given an initial unentangled product state in the computational basis, the associated quantum drums get fixed and come in a variety of shapes and sizes starting from one-plaquette drums to truly extensive

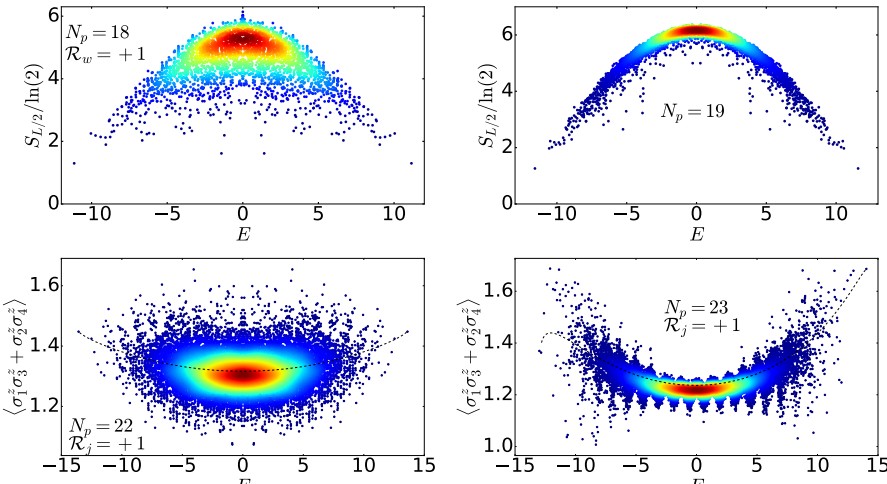

Figure 30: The behavior of the bipartite entanglement entropy for each eigenstate shown for (top left panel) a wire with $N_p = 18$ in the $\mathcal{R}_w = +1$ sector and for (top right panel) a junction of two wires with $N_p = 19$ for both $\mathcal{R}_j = \pm 1$ sectors together. The expectation value of a local diagonal operator defined on an elementary plaquette for each energy eigenstate shown for the wire with $N_p = 22$ (bottom left panel) and the junction of two wires with $N_p = 23$ (bottom right panel) respectively, both in the symmetry sector $\mathcal{R}_{w/j} = +1$. The dotted lines in both the lower panels indicate the thermal values as a function of the energy $E$. In all the panels, the density of states is indicated by the same color map where warmer color corresponds to higher density of states.

structures made of plaquettes that share edges and/or vertices with each other. Specifying the plaquettes that belong to a drum uniquely fixes its spectrum. Crucially, these drums can be "shielded" from each other by shielding regions that only grow as the perimeter and not the area of such drums.

Large quantum drums and their associated fragment dimensions can be most easily estimated by using a "wire" decomposition of such drums and then counting the number of ways in which such wires can fluctuate simultaneously without violating the kinematic constraints. This allows us to identify the appropriate drums that dominate statistically for a given density of up-spins (bosons). The largest Hilbert space fragment is generated by the "checkerboard drum" at a density of $n = 1/4$ for the up-spins (bosons). It is shown that initial states that belong to such fragments evade ETH-predicted thermalization (in the full Hilbert space) due to the presence of either an extensive number of inert spins or an extensive number of next-nearest neighbor spin correlations that retain the memory of the initial state. In particular, initial states that belong to the checkerboard drum fragment contain zero density of inert spins but a finite density of next-nearest neighbor correlations that are pinned to athermal values under time evolution with $H$.

We consider the spectrum of some small drums analytically to show the emergence of interesting zero, non-zero integer and irrational modes. Close packing an extensive number of the elementary one-plaquette drums already generate many-body eigenstates with integer energies (including zero) and strict area-law scaling of entanglement entropy. Large quasi-one-dimensional and two-dimensional quantum drums can be viewed as interesting interacting systems with constrained Hilbert spaces. A class of these drums harbor a large number of exact zero modes. The simplest quasi-one-dimensional drum, which we dub as a wire, is

shown to be exactly equivalent to the well-known PXP chain with open boundary conditions. However, a particular junction of two wires is also studied which cannot be mapped in to the PXP chain and represents a different constrained model. Both these quasi-one-dimensional drums support distinct families of quantum many-body scars that cause periodic revivals from certain simple initial states. Our numerics for the wire also shows that the period-3 state with Rydberg excitations on every third site shows strong revivals for open chains of length $3n+1$ without the necessity of adding further perturbations to the PXP chain. This result can have possible implications for experiments with Rydberg atoms.

Several possible open directions emerge from our study. Other junctions of wires, like junctions of three wires and four wires, as well as some of the two-dimensional drums introduced here should have interesting high-energy properties. It is further possible to add diagonal interactions in the computational basis which preserve the fragmented structure of the model. Using such additional interactions, one can possibly access different phases and phase transitions at zero temperature in both quasi-one-dimensional and two dimensional theories in the presence of subsystem symmetries. Whether many-body localized phases can emerge in quasi one-dimensional and two-dimensional drums in the presence of subsystem symmetries on adding diagonal interactions with random couplings presents another interesting research direction.

*Note added*: While preparing this manuscript, we came to know of a related work by Lehmann *et al.* [83] which discusses strong Hilbert space fragmentation in higher dimensions using a different Hamiltonian (correlated hopping model).

# Acknowledgements

A.C. thanks Madhumita Sarkar for help with cluster facilities at IACS, Kolkata and DST, India for support through SERB project PDF/2021/001134. B.M. has been funded by the European Research Council (ERC) under the European Union's Horizon 2020 research and innovation programme (Grant No. 853368). K.S. thanks DST, India for support through SERB project JCB/2021/000030.

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
