# Peer review of "Strong Hilbert space fragmentation via emergent quantum drums in two dimensions"

_SciPost Physics, doi:SciPost Phys. 14, 146 (2023)_

## Round 2 · Referee Report · Anonymous · 2022-11-30

Strengths
1 - interesting case of non-ergodicity
2- very detailed with analytical results
Weaknesses
1 - unclear whether fragmentation is "true" or "local"
Report
The paper provides and studies in detail a very interesting model of disorder-free localization in 2+1 D. The model hosts disjoint sectors in space that are seemingly responsible for various non-ergodic behavior including fragmentation and scars. The authors call these "quantum drums". Smaller drums admit analytical results and in particular the authors explicitly demonstrate that the model is long-time non-stationary. I find the results of sufficient interest to publish in SciPost Physics.
Requested changes
I am in agreement with the other referee (or comment?) in particular:
1 - with point 5. Can the authors check whether there is any fragmentation beyond the emergent local conservation laws? If not, then the type of fragmentation is called "local" as in https://journals.aps.org/prl/abstract/10.1103/PhysRevLett.128.100601 and this should be discussed.
2- with point 9. The authors breaking of ergodicity is strong because the autocorrelation functions at finite temperature persistently oscillate. This follows from (12) and the proof in https://www.scipost.org/SciPostPhys.9.1.003/pdf The authors should emphasize this point.
>>The paper provides and studies in detail a very interesting model of
>>disorder-free localization in 2+1 D. The model hosts disjoint sectors in space
>>that are seemingly responsible for various non-ergodic behavior including
>>fragmentation and scars. The authors call these "quantum drums". Smaller drums
>>admit analytical results and in particular the authors explicitly demonstrate
>>that the model is long-time non-stationary. I find the results of sufficient
>>interest to publish in SciPost Physics.
Response: We thank the Referee for their careful reading of our manuscript,
for the positive assessment of our work and the questions and comments that
were raised. We respond to all of these below.
>>Weaknesses
>>1 - unclear whether fragmentation is "true" or "local"
Response: Please see detailed response in the next point.
>>1 - with point 5. Can the authors check whether there is any fragmentation
>>beyond the emergent local conservation laws? If not, then the type of
>>fragmentation is called "local" as in
>>https://journals.aps.org/prl/abstract/10.1103/PhysRevLett.128.100601 and this
>>should be discussed.
Response: While it is true that there are certain exponentially large Krylov
subspaces where the Fock states are classically connected to each other but
split up into small blocks when certain linear combinations of a fixed number of
Fock states are used (e.g., see Section 2.3), this only happens for Krylov
subspaces that are made of an extensive number of independent finite-sized
drums. Krylov subspaces obtained from macroscopic 2D quantum drums do not have
such a property as far as we can tell. The macroscopic 2D drums dominate
over an extensive number of independent finite-sized drums owing to
their fragment dimension if the density of up-spins (bosons) $n>1/6$.
>>2- with point 9. The authors breaking of ergodicity is strong because the
>>autocorrelation functions at finite temperature persistently oscillate. This
>>follows from (12) and the proof in
>>https://www.scipost.org/SciPostPhys.9.1.003/pdf
>>The authors should emphasize this point.
Response: Eq. 12 is applicable to the class of initial states that are
consistent with an arrangement of elementary one-plaquette drums and remaining
inert spins. While such Krylov subspaces do contain exponentially large Krylov subspaces
and initial states contained in them do exhibit persistent oscillations, these
are not necessarily the largest Krylov subspaces as a function of the
density of up-spins (bosons) $n$ and thus
strong Hilbert space fragmentation cannot just be argued from the properties of
these fragments alone. For example, the close-packed arrangement of one-plaquette
drums (shown in Fig. 7) gives a fragment size that is
exponentially smaller than the 2D drum shown in Fig. 13 (panel A), even though
both fragments generate initial states with the same density $n=2/9$.
The analytic arguments for strong Hilbert space fragmentation are given in
Section 3.3 in detail (please also see our reply to Anonymous [id 3085]
regarding point 9 raised by them).
Anonymous on 2022-11-29 [id 3085]
This work provides a very detailed analysis of a spin-1/2 2d dimensional system realizing the phenomenon of Hilbert space fragmentation, as well as other phenomena related to the spectral and dynamical properties of this system, as e.g., quantum many-body scars. In particular, the authors argue that the Hilbert space decomposes as a direct sum of dynamically decoupled subspaces that can be labeled in terms of quantum drum configurations. In the following, I share some questions and comments that I hope can contribute to improving this already nice work.
Anonymous on 2023-01-26 [id 3274]
(in reply to Anonymous Comment on 2022-11-29 [id 3085])>>This work provides a very detailed analysis of a spin-1/2 2d dimensional system realizing the >>phenomenon of Hilbert space fragmentation, as well as other phenomena related to the spectral >>and dynamical properties of this system, as e.g., quantum many-body scars. In particular, the >>authors argue that the Hilbert space decomposes as a direct sum of dynamically decoupled
>>subspaces that can be labeled in terms of quantum drum configurations. In the following, I share >>some questions and comments that I hope can contribute to improving this already nice work.
Response: We are grateful to the anonymous commenter for the careful reading of our manuscript, for liking our work, and for the questions and comments that were raised. We respond to the comments below.
>>1. In case it is true, I would explicitly write that the Hilbert space can be decomposed as a direct sum >>over dynamically disconnected sectors that are completely labeled by drum configurations.
Response: This statement is indeed true and was already stated in Section 2.1 of the earlier version---"All many-body eigenstates of $H$...that do not belong to any quantum drum." We have now added an additional line in the Introduction to make this point even more explicit.
>>2. The authors could extend the generality of their results by allowing the coefficient "J" in Eq.(1) >>to vary in space (i.e., J_{jx,jy}) , since this does not affect the results about Hilbert space >>fragmentation.
Response: Thanks for this useful comment. We have now included a paragraph just before Section 2.1 to highlight this and also note that an arbitrary $J(j_x,j_y)$ would still preserve the $E$ to $-E$ symmetry of the spectrum but would greatly reduce the number of non-trivial zero modes emerging from fragments of sizes greater than one.
>>3. In a similar spirit, I would suggest the authors to differentiate those properties listed in page 6 >>that do not hold in the presence of additional diagonal (in the local z-basis)--- and hence not >>affecting the fragmented structure--- contributions. For example, the anticommuting symmetry
>>C in Eq.(4) is only present in the absence of such terms.
Response: We have also included a discussion of this point in the same paragraph that has been inserted just before Section 2.1.
>>4. It is not clear what "exact non-zero integer eigenstates" means in page 7, although one can >>guess what is meant later in the text.
Response: We have now clarified this by adding the phrase "when the Hamiltonian has the normalization of $J=1$ in Eq. 1."
>>5. Did the authors check or can argue whether apart from the subsystem symmetries, other local >>conserved quantities exist when including the constraints appearing in Eq. (2)? The fact that the >>authors additionally constrain an already very constrained system realizing subsystem >>symmetries, could lead to such local conserved quantities. Alternatively, can the authors
>>identify the different quantum drum configurations alabeleded by the eigenvalues of a conserved >>quantity similarly to what has been done for certain 1D systems?
Response: We do not observe any other local conserved quantities apart from ones like total magnetization (for all drums) and certain point-symmetries like reflections (for certain drums depending on their shapes). While it is true that certain exponentially large Krylov subspaces have additional algebraic structures like emergent dynamical symmetry (See Eq. 12 in Section
2.3), this is because such fragments are decomposible into an extensive number of independent finite-sized drums. We do not expect such additional conservations in generic 2D drums like checkerboard drum and close packed drum (Fig. 13). One numerical evidence for it is shown in
Fig. 23 (Section 5.2) where the level statistics of the reflection-symmetry resolved sector of two quasi-1D drums seem to follow GOE.
>>6. On page 11, the authors provide two very detailed examples for obtaining the shielding regions. >>While it serves a pedagogical purpose, I miss a general proof showing that indeed "shielding >>regions" do shield the quantum drum configuration in general.
Response: We are thankful for this comment and have now rewritten Sec. 2.1.2 a bit so that it becomes clear that such "shielding regions" do shield any two finite-sized drums from each other.
>>7. I think that the authors use two different conventions to denote the size of fragments. >>Sometimes they use the dimension of the relevant Hilbert space, and others they use the >>dimension of the matrix representation. See for example page 15.
Response: We have now used the dimension of the relevant Hilbert space throughout the text.
>>8. The notation and discussion about different types of drum configurations appearing in page 16 is not easy to follow. Perhaps the authors can briefly introduce the notation, or light a bit the discussion. It might help to have Fig.6 in the same page.
Response: We have now rewritten Sec. 2.2 a bit to shorten it and also simplify the discussion.
>>9. I think it should be made clear in the discussion and in the introduction, that the strong >>fragmentation of the Hilbert space is only supported by the numerical results contained in Section >>3.1.
Response: We are sorry to hear that our presentation in the previous version gave this impression. The strong Hilbert space fragmentation of this model is not only supported by numerical results in Section 3.1 but also by analytic arguments that use a "wire decomposition" of quantum drums to construct the largest Krylov subspaces for a given density of up-spins (bosons), $n$, to show
that typical initial states possess either an extensive number of inert spins or an extensive number of next-nearest neighbor spin correlators that are pinned to their initial athermal values.
We have now tried to arrange the analytic arguments given in the previous version to make the chain of arguments more explicit. We have moved Section 3.2 of the previous version to Section 2.4 now. This Section is now broken into two further subsections 2.4.1 and 2.4.2 to show (a) the calculation of the fragment dimension from wire decomposition and (b) to construct entire drums from "wire-decomposed reference states" using some examples of small drums to make it apparent that non-wire drums can be generated by coupling parallel wires of unequal lengths. The discussion for macroscopic drums and how to use wire decomposition to estimate their fragment sizes is mainly explained in Section 3.3. In Section 3.3, the drums with the largest Krylov subspaces are
constructed as a function of $n$ and then athermal local operators are identified in typical initial states from such large fragments to show that these initial states do not thermalize with respect to the full Hilbert space, unlike in a situation with weak Hilbert space fragmentation. We have inserted
some additional clarifying remarks to sections 3.2 and 3.3 (as well as Section 2.4) to present the analytic arguments in a clearer fashion.
Furthermore, in Fig. 11 (right panel), we have inserted a curve $(\varphi^{1/4}/\kappa)^N$, where $\varphi$ is the golden ratio and $\kappa$ is the hard square entropy constant, and show that there is a very good match with the ED data for the ratio of the largest fragment size to the total Hilbert space dimension at $n=1/4$ as expected from our theoretical picture (explained in Section 3.3). We have inserted a new Figure 15 in Section 3.3. to summarize which drums dominate statistically as a
function of $n$. We have also made it explicit that the " 2D checkerboard drum" produces the largest Hilbert space fragment for this model with initial states in this fragment containing a zero density of inert spins but an extensive density of athermal next-nearest neighbor spin correlations.

---

## Round 2 · Referee Report · Anonymous · 2022-12-6

Strengths
1. An interesting model display Hilbert space fragmentation in 2D.
2. Detailed analysis of the emergent quantum drum and its dynamics, and special attention is given to drums of wire structures.
3. A combination of analytical and numerical results.
Weaknesses
1. The evidence of strong fragmentation seems to be only numerical, mainly because it is unknown what the largest Hilbert space is beyond that contains only wires.
2. The model conserves particle numbers in each column and row. It is unclear whether this extensive amount of conservation laws is required for the fragmentation and drum.
3. It is unclear whether the discussion in 3.4 is about thermalization or thermalization in the Krylov subspace.
4. The paper contains too many details, and it is easy to get lost in them and miss the main point.
Report
This work provides a rare model displaying fragmentation in 2D and complex emergent structures dubbed drums. The fragmentation is made possible by supplementing ring exchange interaction with additional nearest neighboring Rydberg blockade kinetic constraint. The authors analyze the model and the drums in detail, especially those made of wires. They show that wire drums can be mapped to the 1D PXP model and display non-ergodic dynamics due to many-body scars. I believe that this work is suitable for publication in scipost.
Requested changes
1. The authors should be clear on what is done to demonstrate strong fragmentation in the introduction. If I understand it correctly, the evidence is numerical, and we don't know how to construct the largest Hilbert space sector.
2. The authors should comment on the effects of charge conservation in each column and each row. Is it necessary for strong fragmentation?
3. It is unclear whether the discussion in 3.4 is about thermalization or thermalization in the Krylov subspace. In systems with fragmentation, it is not surprising that thermalization is broken because the Hilbert space is disconnected, and what is relevant is thermalization within each sector. Even though an initial state in the computational basis has energy 0, it does not mean that an observable should thermalize to its trivial trace. The state and its chiral partner might not be connected by the Hamiltonian. The authors should clarify this point in 3.4.
>>Weaknesses
>>1. The evidence of strong fragmentation seems to be only numerical, mainly
>>because it is unknown what the largest Hilbert space is beyond that contains
>>only wires.
Response: Please see detailed response to a related point later in our response.
The concept of wire-decomposition used in our work shows how to create non-drum wires by coupling parallel wires, with unequal lengths in general. Furthermore, certain simplifications emerge
for macroscopic drums composed of long wires using which the scaling of the corresponding fragment size can be worked out.
>>2. The model conserves particle numbers in each column and row. It is unclear
>>whether this extensive amount of conservation laws is required for the
>>fragmentation and drum.
Response: While we believe that fragmentation can arise even without these
subsystem symmetries, combining subsystem symmetries with a constrained
Hilbert space allowed us to have a model with (a) emergent quantum drums and
(b) strong fragmentation. In particular, models with
strong fragmentation are hard to construct in 2D as far as we understand.
>>3. It is unclear whether the discussion in 3.4 is about thermalization or
>>thermalization in the Krylov subspace.
Response: Please see the detailed reply to this point and a related question later.
>>4. The paper contains too many details, and it is easy to get lost in them and
>>miss the main point.
Response: We have now tried to improve the presentation of the paper by
adding some clarifying remarks at appropriate places. We have also
rearranged some of the sections (see a detailed
response later) for ease of reading.
>>This work provides a rare model displaying fragmentation in 2D and complex
>>emergent structures dubbed drums. The fragmentation is made possible by
>>supplementing ring exchange interaction with additional nearest neighboring
>>Rydberg blockade kinetic constraint. The authors analyze the model and the
>>drums in detail, especially those made of wires. They show that wire drums can
>>be mapped to the 1D PXP model and display non-ergodic dynamics due to
>>many-body scars. I believe that this work is suitable for publication in
>>scipost.
Response: We thank the Referee for their careful reading of our manuscript,
for the positive assessment of our work and the questions and comments that
were raised. We respond to all of these below.
>>1. The authors should be clear on what is done to demonstrate strong
>>fragmentation in the introduction. If I understand it correctly, the evidence
>>is numerical, and we don't know how to construct the largest Hilbert space
>>sector.
Response: The strong Hilbert space fragmentation of this model is not only
supported by numerical results in Section 3.1 but also by analytic arguments
that use a "wire decomposition" of quantum drums to construct the largest
Krylov subspaces for a given density of up-spins (bosons), $n$, to show
that typical initial states possess either an extensive number of inert spins
or an extensive number of next-nearest neighbor spin correlators that are
pinned to their initial athermal values.
Motivated by the Referee's comment, we have now tried to arrange the analytic
arguments given in the previous version to make the chain of arguments more
explicit. We have moved Section 3.2 of the previous version to Section 2.4
now. This Section is now broken into two further subsections
2.4.1 and 2.4.2 to show (a) the calculation of the fragment dimension from
wire decomposition and (b) to construct entire drums from "wire-decomposed
reference states" using some examples of small drums to make it apparent
that non-wire drums can be generated by coupling parallel wires of unequal
lengths. The discussion for macroscopic drums and how to use wire
decomposition to estimate their fragment sizes is mainly explained in
Section 3.3. In Section 3.3, the drums with the largest Krylov subspaces are
constructed as a function of $n$ and then athermal local operators are
identified in typical initial states from such large fragments to show that
these initial states do not thermalize with respect to the full Hilbert space,
unlike in a situation with weak Hilbert space fragmentation. We have inserted
some additional clarifying remarks to
sections 3.2 and 3.3 (as well as Section 2.4) to present the analytic arguments
in a clearer fashion.
Furthermore, in Fig. 11 (right panel), we have inserted a curve
$(\varphi^{1/4}/\kappa)^N$, where $\varphi$ is the golden ratio and
$\kappa$ is the hard square entropy constant, and show that there is a very
good match with the ED data for the ratio of the largest fragment size to the
total Hilbert space dimension at $n=1/4$ as expected from our theoretical
picture (explained in Section 3.3). We have inserted a new Figure 15
in Section 3.3. to summarize which drums dominate statistically as a
function of $n$. We have also made it explicit that the " 2D checkerboard
drum" produces the largest Hilbert space fragment for this model with
initial states in this fragment containing a zero density of inert spins but
an extensive density of athermal next-nearest neighbor spin correlations.
We hope that the analytic arguments are now clear from our improved presentation.
>>2. The authors should comment on the effects of charge conservation in each
>>column and each row. Is it necessary for strong fragmentation?
Response: In this model on the square lattice, the charge conservation on
each column and row leads to a global dipole moment conservation and hence
Hilbert space fragmentation as was already noted in Ref[70].
However, we need the further ingredient of hard-core constraints in the
Hilbert space (Eq. 2) to observe strong fragmentation as well as the
emergent quantum drums. These points are already mentioned in the Introduction.
>>3. It is unclear whether the discussion in 3.4 is about thermalization or
>>thermalization in the Krylov subspace. In systems with fragmentation, it is
>>not surprising that thermalization is broken because the Hilbert space is
>>disconnected, and what is relevant is thermalization within each sector. Even
>>though an initial state in the computational basis has energy 0, it does not
>>mean that an observable should thermalize to its trivial trace. The state and
>>its chiral partner might not be connected by the Hamiltonian. The authors
>>should clarify this point in 3.4.
Response: We thank the Referee for this comment and have now added some
clarifying remarks in Section 3.3 (Section 3.4 of the earlier version). It is
indeed true that talking about initial states in the computational basis is
perhaps slightly confusing here. We have now changed that phrasing to "typical
initial states". Since all fragments of this model have $E$ to -$E$ symmetry,
typical initial states from any exponentially large Krylov subspace
are expected to always thermalize to the infinite
temperature ensemble of that subspace (Krylov-restricted ETH). However, the
crucial point is to establish that there is no single dominant Krylov subspace
in the thermodynamic limit to justify strong fragmentation. We establish
precisely this in Section 3.3 for typical initial states in the largest
Krylov subspaces by showing that certain local spin correlations stay
pinned to their initial athermal values (with respect to the full Hilbert space), thus
ruling out a dominant Krylov subspace (please also see our reply to
Anonymous [id 3085] regarding a related point 9 raised by them). For typical
initial states, thermalization in the full Hilbert space does imply a
simple trace over the entire Hilbert space and we show that this is not the
case in Section 3.3. We have also changed the title of Sec. 3.3 to "Large Krylov
subspaces and absence of ETH-predicted thermalization" from
Absence of thermalization from typical product states".

---

## Round 3 · Referee Report · Anonymous · 2023-1-26

Report

I have reviewed the revised version, and my concerns have been addressed. I can recommend publication in scipost.

---

## Round 3 · Referee Report · Anonymous · 2023-1-30

Report

The authors have addressed my remarks and I recommend publication.

---

## Round 3 · Author Response

Dear Editor,

Thanks for sending us the referee comments and also the comments by an anonymous commenter. The questions and comments were very useful and helped us improve the presentation of the paper and also remove some points of confusion. We have now responded to all the comments from both referees and the anonymous commenter (in the form of a reply to their individual reports) and hope that you would find the present version to be publishable in SciPost
Physics.

With best regards,

Anwesha Chattopadhyay, Bhaskar Mukherjee, K. Sengupta, Arnab Sen

---

## Round 3 · List of Changes

1. Changed a phrase in the Abstract.

2. Added a sentence to the introduction on the suggestion of the anonymous commenter.

3. Added a couple of phrases to the introduction to clarify a few points that the referees had found confusing.

4. Added a phrase and a paragraph to Section 2 (before the beginning of Section 2.1) on the suggestion of the anonymous commenter.

5. Section 2.1.2 has been rewritten a bit to clarify a point raised by the anonymous commenter.

6. Section 2.2 has been shortened a bit and the notation simplified on the suggestion of the anonymous commenter.

7. The earlier Section 3.2 titled "Wire decomposition of quantum drums" has now been moved to Section 2.4 which consists of two subsections titled Sec. 2.4.1 "Calculating fragment dimension from wire decomposition" and Section 2.4.2 "Constructing entire drums from wire-decomposed reference states". This rearrangement and some clarifying remarks here should make it easier to follow some of the analytic arguments.

8. Some clarifying remarks have been added in Section 3 to make the analytic arguments for strong Hilbert space fragmentation more explicit after receiving the referee reports. We have also changed the title of Sec. 3.3 to
"Large Krylov subspaces and absence of ETH-predicted thermalization" from Absence of thermalization from typical product states" (Sec. 3.4 of the previous version).

9. An analytic prediction in the form of a dotted curve has been added to Fig. 11 (right panel).

10. A new schematic figure (Fig. 15) has been added to summarize the fate of a typical initial state as a function of up-spins (bosons), $n$.

11. Added a couple of sentences in the section: Discussion to highlight a few points mentioned by the referees.

---

## Editorial Decision

published